# Direct coupling of light to valley current

S. Sharma ®[1,2] ✉, D. Gill ®[1], J. Krishna ®[1], J. K. Dewhurst[3] & S. Shallcross ®[1] ✉

The coupling of circularly polarized light to local band structure extrema ("valleys") in two dimensional semiconductors promises a new electronics based on the valley degree of freedom. Such pulses, however, couple only to valley charge and not to the valley current, precluding lightwave manipulation of this second vital element of valleytronic devices. Contradicting this established wisdom, we show that the few cycle limit of circularly polarized light is imbued with an emergent vectorial character that allows direct coupling to the valley current. The underlying physical mechanism involves the emergence of a momentum space valley dipole, the orientation and magnitude of which allows complete control over the direction and magnitude of the valley current. We demonstrate this effect via minimal tight-binding models both for the visible spectrum gaps of the transition metal dichalcogenides (generation time ~ 1 fs) as well as the infrared gaps of biased bilayer graphene (~ 14 fs); we further verify our findings with state-of-the-art time-dependent density functional theory incorporating transient excitonic effects. Our findings both mark a striking example of emergent physics in the ultrafast limit of light-matter coupling, as well as allowing the creation of valley currents on time scales that challenge quantum decoherence in matter.

Novel quantum degrees of freedom of the solid state, such as valley and topological charge, represent rich avenues for the ultrafast control of matter by light while, at the same time, providing a route towards future classical and quantum computing architectures[1]. The valley degree of freedom, that directly couples to circularly polarized light, has in particular attracted attention, both as a potential "qubit"[2–4] as well as for the diverse free carrier and excitonic valley physics observed in the transition metal dichalcogenide family[3,5–11]. Circularly polarized light can, however, only address the valley charge state. Thus while such pulses offer important control over the state of the "information bit" of valleytronics they cannot generate valley current[12,13], an essential ingredient of any future spintronic or valleytronic devices.

This absence of current control would appear fundamental, as to generate ultrafast pure valley current apparently necessitates the combination of two contradictory light pulses: a waveform possessing a fixed polarization vector to generate current by distinguishing directions in momentum space[14–17], as well as a rotating polarization vector to exclusively couple light to a single valley. Stated in another way, as a waveform characterized by a scalar, its helicity, it might be expected that circularly polarized light will couple only to scalar degrees of freedom, such as the valley charge state, and not to vector valued observables such as the valley current.

More complex pulse designs, such as augmenting a circularly polarized pulse by linearly polarized light (a hybrid "hencomb" pulse[18]) lifts this restriction, however such pulses are intrinsically long time hybrids as the linearly polarized component, designed to induce only intra-band transitions, requires a substantially below-gap frequency[18]. It would therefore appear that the generation and control of valley current on the all-important few femtosecond time scale, essential for valleytronic challenges to quantum decoherence, is not possible.

Surprisingly, this "common sense" conclusion turns out to be false: we show that the ultrafast few cycle limit of circularly polarized light breaks valley point group symmetry, physically manifest as a high purity valley current. The underlying symmetry breaking mechanism involves the emergence of a dipole structure to the excited charge in momentum space (a "K-pole") whose magnitude and direction, determined by the pulse, completely controls valley current. This has

[1]Max-Born-Institut für Nichtlineare Optik und Kurzzeitspektroskopie, Max-Born-Strasse 2A, 12489 Berlin, Germany. [2]Institute for theoretical solid-state physics, Freie Universität Berlin, Arnimallee 14, 14195 Berlin, Germany. [3]Max-Planck-Institut für Mikrostrukturphysik Weinberg 2, D-06120 Halle, Germany. ✉e-mail: geet1729@gmail.com; phsss75@gmail.com

its origin in the dependence of the dynamics of the interband dipole matrix element on the "shape" of the light induced evolution of crystal momentum $\mathbf{k}(t)$, and the fact that the many and single cycle limits of circularly polarized light possess distinct forms of $\mathbf{k}(t)$—approximately circular and a low symmetry single loop respectively. We argue such symmetry breaking characterizes a "borderland" regime between the "strong field" attosecond limit, much explored in atomic physics, and regime of multicycle light, much explored in the solid state. We demonstrate this physical picture holds for the full range of gap sizes relevant for valleytronics: from the visible spectrum gaps of transition metal dichalcogenides, to the infrared gaps that can be created in bilayer graphene[19,20]. We further demonstrate, employing a recent Kohn-Sham-Proca scheme[21] capable of treating ab-initio excitons both in linear response and in highly non-linear laser pump regime, that symmetry breaking mechanism is robust to the inclusion of excitonic effects. Our findings both mark a notable example of the fundamental differences that can occur between short time and long time manifestations of the same light-matter phenomena, as well as establishing the fastest possible mechanism in which valley polarized current and charge can be created and controlled in a wide range of systems.

## Results

### Fast is different: valley symmetry breaking by circularly polarized light

We first explore emergent symmetry breaking of light-valley coupling in the visible gap regime. To this end we employ a by now standard model of the valley active transition metal dichalcogenide family, the graphene Hamiltonian with a mass term ("gapped graphene")[5], with time dependence and phenomenological quantum decoherence treated via propagation of the density matrix, see "Methods" for elaboration of this technique. In recent work we have bench-marked this approach against state-of-the-art ab-initio time-dependent density functional theory calculations for a range of pulse durations, finding good agreement[11,18]. This approach thus represents an efficient and predictive tool for investigating light-matter interaction in valley active materials.

A multi-cycle circularly polarized pulse excites valley charge but not valley current. In Fig. 1a we show the vector potential of a "typical" multicycle pulse (red light central frequency $\omega = 1.6$ eV, full width half maxima 23.5 fs, and amplitude $A_0 = 2.05$ a.u.). A pronounced charge excitation, valley polarized at K, panels (b,c), is accompanied by the vanishing of post-pulse current, panels (e). As we now show, this "standard model" of circularly polarized light fails in the single cycle limit. For a 2.4 fs duration pulse ($A_0 = 6.85$ a.u.), panel (f), the resulting dynamics creates not only a valley polarized charge excitation, panels (g-i), but also a significant valley current density excitation, with a post-pulse residual of 3.27 $\mu$A/nm, panel (j). Note that for analyzing residual current density we show the intraband contribution; the full current density, see Supplemental Sec. 1, behaves in exactly the same way but with the presence of the usual decaying transient oscillation. The ultrafast limit of light-valley coupling is, we must conclude, associated with the emergence of qualitatively new physics not found at longer times.

A short time emergent phenomena implies an underlying laser induced symmetry breaking. Surprisingly, however, comparison of the momentum resolved conduction band occupation—panels (b,c) and (g,h)—reveals no "obvious" difference between the multi-cycle and single-cycle pulses. To unveil the "hidden" symmetry breaking we define the density:

$$D(\mathbf{k}) = |c_{\mathbf{k}}|^2 - \frac{1}{3}\sum_{i=1}^{3} |c_{M_i\mathbf{k}}|^2, \qquad (1)$$

which encodes the subtraction from the conduction band occupation at $\mathbf{k}$, $|c_{\mathbf{k}}|^2$, the average of the conduction band occupations over the 3 $\mathbf{k}$-vectors related by the valley $C_3$ symmetry operations $M_i$ (i.e., the star average of excited charge). A laser induced charge excitation exactly respecting valley symmetry will have $D(\mathbf{k}) = 0$ for all $\mathbf{k}$. Concomitantly, finite values of $D(\mathbf{k})$ indicate a breaking of valley symmetry. The density $D(\mathbf{k})$ thus represents a "symmetry breaking density", measuring momentum resolved lowering of local valley symmetry in the charge excitation.

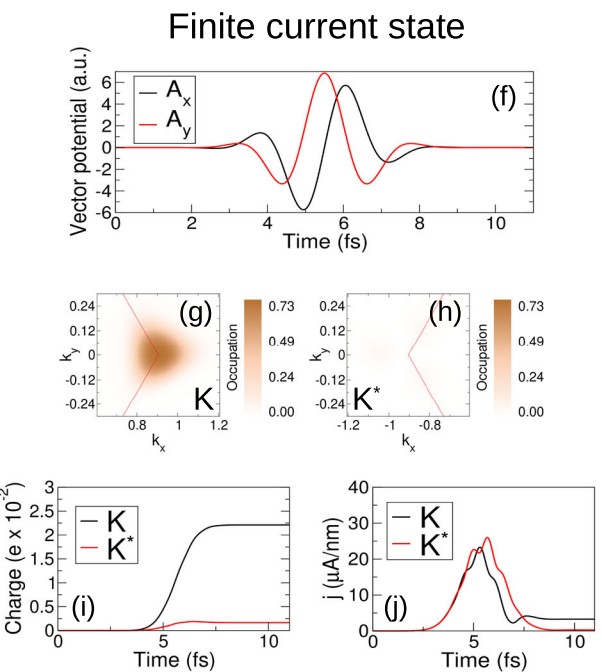

**Fig. 1 | Emergent properties of circularly polarized light in the single cycle limit.** Long time multicycle (23.5 fs duration, panel (**a**)) and ultrashort single cycle (2.3 fs, panel (**f**)) pulses generate very similar valley charge excitations, with a pronounced excitation at the K valley but almost no excitation at the conjugate K' valley, panels (**b**, **c**) and (**g**, **h**), respectively. The current density responses are, however, dramatically in contrast: while the long time pulse yields the zero valley current density state expected of circularly polarized light, the single cycle pulse generates a finite and large valley current density, compare panels (**e**, **j**).

## Zero current state

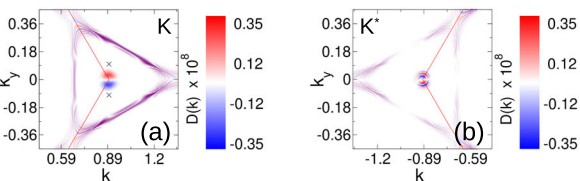

## Finite current state

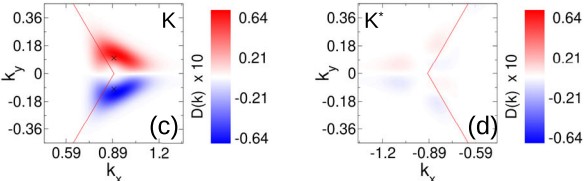

**Fig. 2 | "K-pole" dipole moment characterizing the emergence of an early time symmetry breaking regime of valley coupling by circularly polarized light. a, b** For long time pulses $C_3$ valley symmetry is preserved in the laser induced charge excitation and the symmetry breaking density $D(\mathbf{k})$, Eq. (1), is therefore negligible. At pulse durations of ~1–2 fs light-matter coupling is driven to a

symmetry breaking regime, (**c, d**), with the laser induced charge excitation exhibiting a finite "K-pole" structure to the momentum resolved excited state population with clear $+k_y \rightarrow -k_y$ mirror symmetry breaking. In this regime the lightform generates a valley current.

With this tool in hand we can proceed to analyze the symmetry properties of the multi-cycle and single-cycle charge excitations of Fig. 1. As expected, the long time pulse represents a high symmetry situation, generating a charge excitation for which $D(\mathbf{k})$ ~$10^{-8}$ at both valleys, Fig. 2a, b. In striking contrast, in the single cycle limit the local $C_3$ valley symmetry is clearly broken, with the K valley revealing the emergence of a distinct "dipole-like" structure to the $D(\mathbf{k})$ density, panels (c,d). As a charge excitation obeying $C_3$ symmetry implies zero current—the Bloch velocities $\nabla_{\mathbf{k}}\varepsilon(\mathbf{k})$ will cancel when summed over each set of $C_3$ related $\mathbf{k}$-vectors—the $D(\mathbf{k})$ density, directly measuring the deviation of the conduction band occupation from this $C_3$ symmetry, yields immediate insight into the origin of short time valley currents. For the current carrying single cycle excitation, Fig. 2c, $D(\mathbf{k})$ reveals reduction of conduction band occupation for $k_y < 0$ with corresponding increase at $k_y > 0$, implying a valley current in the $+y$ direction. This corresponds exactly to the emergent current seen in Fig. 1(f–j), confirming the correctness of this interpretation of the laser induced dynamics.

We now consider the physical mechanism underpinning this $k_y \rightarrow -k_y$ symmetry breaking exhibited in Fig. 2c, d and the associated current generation. For the multicycle pulse the trajectory determined by the Bloch acceleration theorem, $\mathbf{k}(t) = \mathbf{k}(0) - \mathbf{A}(t)/c$, is shown in Fig. 3a for two initial momenta $\mathbf{k}(0) = (0, \pm k_y)$, indicated by the crosses, presented along with the normalized magnitude of the interband dipole matrix element $d(\mathbf{k})/d(\mathbf{0})$ that determines the interband excitation, see Supplemental Section 2. Many cycle circularly polarized pulses execute an approximately circular motion in momentum space, a situation that, as may be seen in Fig. 3a, leads to an approximate mirror $k_y \rightarrow -k_y$ symmetry of the trajectories for the two initial momenta $(0, \pm k_y)$. As $d(\mathbf{k})$ also respects $k_y \rightarrow -k_y$, the "recorded" magnitude of the interband dipole matrix element by each trajectory will thus be very similar. This can be seen in Fig. 3b in which the $d(\mathbf{k})$ is plotted as a function of time for each initial momenta. The oscillation results from the two initial momenta $(0, \pm k_y)$ successively evolving closer to the region of high $d(\mathbf{k})$ at K and subsequently away from it, while the phase shift has its origin in the fact that while the $(0, +k_y)$ trajectory evolves towards K the $(0, -k_y)$ trajectory necessarily evolves away from K. Regions of high interband dipole matrix elements are associated with excitation of charge to the conduction band, however over many cycles the effect for the two initial momenta averages out and, as a result and as may be seen in Fig. 3c, the conduction band occupations post laser pulse are nearly identical, $|c_{+k_y}|^2 \sim |c_{-k_y}|^2$.

For the single cycle pulse, Fig. 3d–f, the situation is very different. Now the $\mathbf{k}(t)$ trajectory executes essentially only one loop, which may either be *into* the region of high interband dipole matrix element at K, or *away* from it, see Fig. 3d. For the $(0, +k_y)$ initial momenta, light evolution of momenta into high $d(\mathbf{k})$ will result in increased excitation to the conduction band as compared to the $(0, -k_y)$ initial condition, that evolves away from this region. This may be seen in Fig. 3e, f. We

now have $|c_{+k_y}|^2 > |c_{-k_y}|^2$ and the near equality of the conduction band occupations found for the multi-cycle case has been broken. Similarly, for all valley momenta $k_y > 0$ the charge excited to the conduction band will be greater than that for valley momenta $k_y < 0$, lowering the symmetry of the charge excitation from that of the underlying $C_3$ symmetric valley manifold, and manifesting as the K-pole structure shown in Fig. 2c, d and the associated valley current.

This insight into the underlying dynamics also resolves the "degree of freedom" conundrum for circularly polarized pulses: how can a pulse characterized by a scalar—the helicity—couple to a vector valued observable, the valley current? Long time pulses are characterized by a uniformly rotating polarization vector, with the sign of this rotation the key (scalar) physical property of the pulse. In contrast, at the few limit the electromagnetic potential becomes a single loop that, while inheriting the crucial sign of rotation from its long time counterpart, is now *orientable* implying emergent vectorial character. This key difference can immediately seen in the $\mathbf{k}(t)$ dynamical trajectories in Fig. 3a, d. For the long duration pulse $\mathbf{k}(t)$ possesses approximate circular symmetry, exhibits no preferred direction in momentum space, and cannot therefore generate current. However upon reduction to a single cycle $\mathbf{k}(t)$ clearly breaks mirror symmetry, and the orientable $\mathbf{k}(t)$ can now generate current.

### Controlling valley currents

The ultimate goal of any scheme of valley current generation is complete control, via lightwave form, over both the magnitude and direction of the induced current. In the single cycle limit the carrier envelope phase $\phi_g$, that determines the orientation angle of the $\mathbf{A}(t)$-trajectory in momentum space, emerges as a physically relevant variable, suggesting that this expanded set of pulse parameters can be used to control valley current. The definition of the carrier envelope phase $\phi_g$ can be found in "Methods", see Eq. (8).

In Fig. 4a–c we display three single cycle waveforms with distinct $\phi_g$, along with the complete carrier envelope phase (CEP) dependence of valley current $(J_x(\phi_g), J_y(\phi_g))$, shown in Fig. 4g. The evident sinusoidal variation indicates perfect directional control via the $\phi_g$. Changes in $\phi_g$ are reflected directly as a corresponding rotation of the underlying "dipole like" structure of the symmetry breaking density $D(\mathbf{k})$, Fig. 4d–f, identified in the preceeding discussion as key to the emergence of current at ultrashort times. In analogy to a real space dipole, we may define a dipole moment of this momentum space valley "K-pole": $\mathbf{P} = \sum_{\mathbf{k}} D(\mathbf{k})\mathbf{k}$. In Fig. 4g is shown this moment $\mathbf{P}$ revealing as expected a valley current is directly proportional to $\mathbf{P}$ (a fact that follows directly from the linearity of $\nabla_{\mathbf{p}}H(\mathbf{p})$ close to the K point). Furthermore, increasing the magnitude of the pulse vector potential, Fig. 4h, leads to increase in current with the "K-pole" moment magnitude exactly in step. The valley "K-pole" moment thus represents the fundamental emergent physical property of the symmetry breaking regime of circularly polarized light. Note that here we have employed

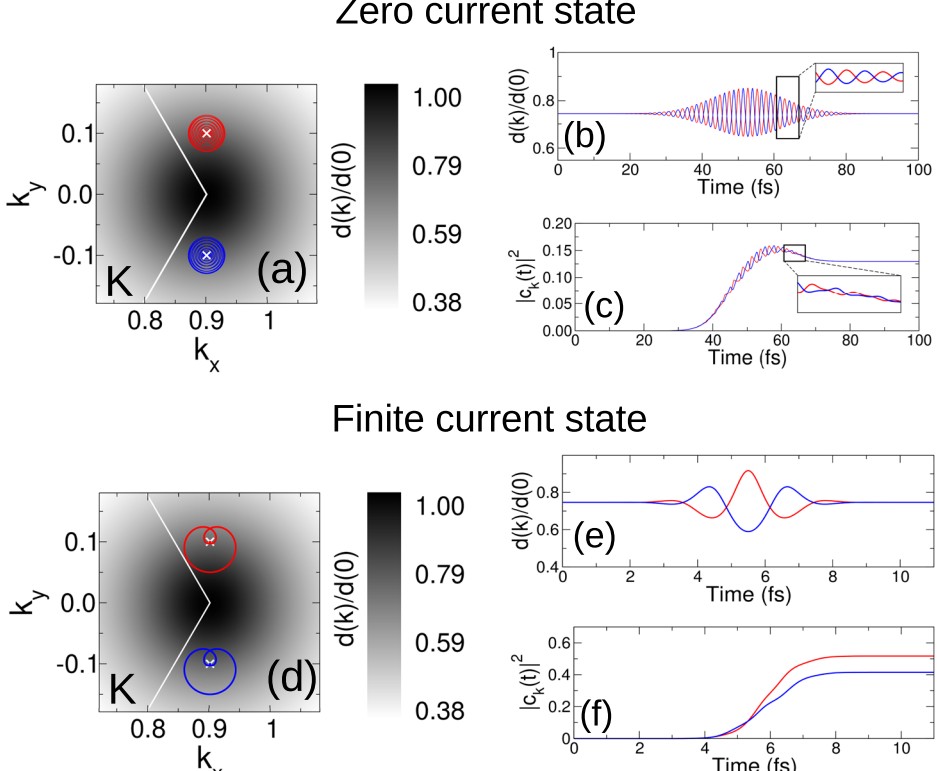

**Fig. 3 | The physical mechanism of emergent short time symmetry breaking.**
**a** For long time pulses, the laser driven $\mathbf{k}(t)$-trajectories for two initial momenta $(0, +k_y)$ and $(0, -k_y)$, indicated by the crosses in panel (**a**), approximately map into each other under $k_y \to -k_y$ and, as a result, the (normed) magnitude of the interband dipole matrix element "recorded" along the two dynamical trajectories, $d(\mathbf{k}(t))$, exhibit similar but phase shifted oscillations, panel (**b**), generating ultimately nearly identical final $|c_\mathbf{k}|^2$, panel (**c**). Reduction in pulse duration to ~1–2 fs drives light-matter coupling to a symmetry breaking regime, (**d–f**). The few cycle laser driven

trajectories for initial momenta $(0, \pm k_y)$ now lack $k_y \to -k_y$ reflection symmetry, panel (**d**), with, as a consequence, the "recorded" $d(\mathbf{k})/d(\mathbf{0})$ along the two trajectories now very different, panel (**e**). The $+k_y$ initial momenta—whose dynamical $\mathbf{k}(t)$ trajectory drives it *into* the region of high $d(\mathbf{k})$—has increased conduction band charge $|c_\mathbf{k}|^2$ as compared to the $-k_y$ initial momenta whose dynamical trajectory drives it *away* from the region of high $d(\mathbf{k})$ at K, (**f**). The laser induced charge excitation inherits this symmetry breaking with a finite "K-pole" structure to the momentum resolved excited state population at the K valley, as shown in Fig. 2c, d.

in our analysis the intraband current, which the full current limits to at 20–30 fs time scales. The very short time current, dominated by the interband components, also exhibits the full CEP control possessed by the later time intraband dominated current, see Supplemental Section 1.

### Impact of the Coulomb interaction and excitonic physics

Excitons form an indispensable part of understanding the optical response for materials possessing a band gap. We now therefore address the question of how excitons behave in the symmetry breaking regime of circularly polarized light that we have identified here. In a recent work, Dewhurst et al. have presented a Kohn-Sham-Proca functional for time-dependent density functional theory (TD-DFT)[21] demonstrating that this captures accurately both the linear response limit and, most importantly for our purpose here, the highly non-linear regime of laser pumped dynamics. We will deploy this theory to calculate tungsten diselenide irradiated by a few cycle circularly polarized light pulse, employing the state-of-the-art full-potential linearized augmented plane wave method[22] as implemented in the Elk code[23,24] (see "Methods" for further details).

The absorption spectrum of $WSe_2$ thus calculated can be seen in Fig. 5a showing the excitonic resonance near the gap edge to be very well reproduced in the calculated absorption (we use a scissors correction to match the reduced DFT gap to that of experiment). Pumping this system with a few cycle circularly polarized pulse very similar to that shown in Fig. 1(f), vector potential presented in the Supplemental Section 8, reveals increased current upon inclusion of excitonic

effects, Fig. 5b. The critical test of robustness is whether this current is CEP controllable—the hallmark of an orientable pulse trajectory in momentum space and thus of the symmetry breaking regime. As can be seen in Fig. 5c, d, this clearly so: the sinusoidal oscillation found Fig. 4g is reproduced here in a realistic material, solved ab-initio both in the absence of and with the inclusion of excitonic effects, Fig. 5c, d, respectively. Note that for ease of comparison we have shifted the sinusoidal form by 43.64°, panel (c), and 141.82°, panel (d); as described in the Supplementary Section 1 the full current at short times exhibits a time dependent rotation from the CEP direction. Excitons thus appear to follow the same physics of "K-pole" symmetry breaking as the free carriers, acting to enhance the symmetry breaking current at short times.

### Pulse parameters of the symmetry breaking regime

We now establish the laser pulse parameters that demark the symmetry breaking domain. To this end we create phase diagrams of charge and valley current density response, Fig. 6a–d. These reveal (i) valley coupling fails entirely for sub-single-cycle circularly polarized pulses and (ii) maximum current magnitude falls off as pulse duration is increased or decreased from that of a gap tuned single cycle pulse $h/\Delta$, of the order of 1–2 fs for the 1.6 eV gap we consider here, Fig. 6c. Significant currents can, however, still be generated for few cycle pulses. Most interestingly, comparison of points (A) and (B) indicated on the phase diagrams reveals that tuning of pulse duration and frequency, will allow valley current and charge to be separately controlled by light; a natural consequence of the fact that in the

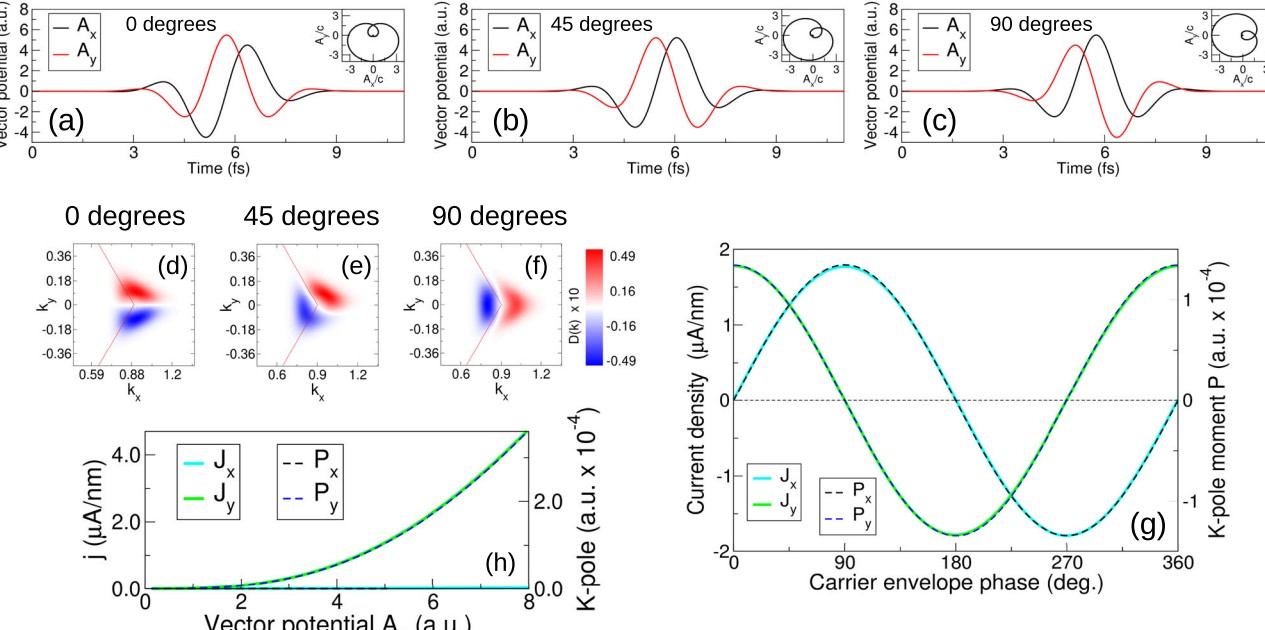

**Fig. 4 | Light control over valley current direction and magnitude.** Ultrashort pulses of circularly polarized light are *orientable* in momentum space, as illustrated by the vector potentials and, inset, their parametric **A**(t)-trajectories, panels (**a**–**c**). This represents an emergent property of circularly polarized light in the few cycle limit, and this orientation can be controlled by the pulse carrier envelope phase (CEP). The "K-pole" structure of the conduction band excitation that characterizes this symmetry breaking regime possesses an orientation exactly equal to the CEP, panels (**d**–**f**). This manifests as a CEP controlled valley current density, panel (**g**). In combination with current density magnitude control via the vector potential amplitude, panel (**h**), complete light control over valley current is achieved. Defining, by analogy with a real space dipole, a "K-pole" dipole moment we find that this, up to a scale constant, is exactly equal to the valley current, highlighting the fundamental role this object plays in the symmetry breaking of light-valley coupling.

emergent symmetry breaking regime the lightform addresses not only valley charge but also the "K-pole" density. Sensible variation of pulse amplitude, see Supplementary Section 4, results only in scaling of valley current magnitudes. The criteria for light induced symmetry breaking is thus gap tuned light of duration $h/\Delta$. To explore this we present explicitly the pulse properties for the three distinct regimes of light: multi-cycle, single cycle, and very short time sub-cycle, indicated by the points (A-C) in the phase diagrams. Each of these regimes is endowed with a distinct pulse form. For the multicycle pulse the parametric trajectory, panel (e), is approximately circular, which upon reducing FWHM first shows a dramatic lowering of mirror symmetry as it becomes effectively a single orientable loop, panel (h): the former belongs to the charge-only coupling regime, while the latter possesses the emergent vectorial character allowing both current- and charge-coupling. At sub-single cycle, (k-m), the pulse trajectory approaches that of linearly polarized light with concomitant reduction in valley polarization, (b) and (d). Moreover, the sharply reduced weight of the pulse Fourier transform close to the gap edge, compare panels (g, j) with (m), acts to suppresses charge and current excitation, as may be observed in the phase diagrams (a) and (c).

### Valley currents for infrared gaps: biased bilayer graphene

Thus far we have considered a minimal model of a valley active material, and we now consider the very different "Mexican hat" valley structure of bilayer graphene that results from the application of interlayer bias to this material[25] (details of the standard tight-binding model we employ and the corresponding low energy band structure are presented in "Methods"). Light-valley coupling in bilayer graphene has recently been investigated in experiment[19], revealing valley charge excitation by pulses of circularly polarized infra-red light. The maximum of the interband dipole matrix element is now localized on the "hat rim", leading to a characteristic annular pattern of conduction

band excitation[20]. Despite this quite different low energy structure, a symmetry breaking current generation is again found, see Fig. 7a, which moreover occurs at the expected pulse duration of $h/\Delta = ~14$ fs. Finally, we confirm that in bilayer graphene complete control over current density magnitude and direction via the pulse carrier envelope phase and amplitude, see Fig. 7b, c, respectively. Once again, the current is seen to follow exactly the "K-pole" moment, reaffirming this as the fundamental emergent property of single cycle valley coupling. Attempts at the creation of valley currents in graphene type materials has a long history[26–30], primarily focused on valley filtering via controlled deformation, however these are yet to be realized in experiment; the approach described here offers a direct route to the ultrafast creation of such currents avoiding the complications of material deformation.

### Discussion

Efforts to control matter by light generally take laser pulse design towards more "complex" waveforms, for instance "hencomb" pulses[18,31]. In contrast, here we show that approaching the single cycle limit of a "simple" waveform—circularly polarized light—unveils a symmetry breaking regime with dramatically enriched light-matter coupling. This regime allows ultrafast control not only over valley charge (a property of "long time" circularly polarized light) but also the creation and control over nearly pure valley and spin currents. While valley charge control points towards the creation of ultrafast classical and, possibly, quantum information bits, dual control with valley current control provides a second crucial ingredient of a future electronics based on the valley degree of freedom.

The physical mechanism is driven by the emergence of a momentum space valley dipole (a "K-pole"), lowering the $C_3$ valley symmetry preserved by longer time laser excitation, with the orientation and magnitude of this dipole directly correlating to the direction and strength of an emergent valley current. The resulting current

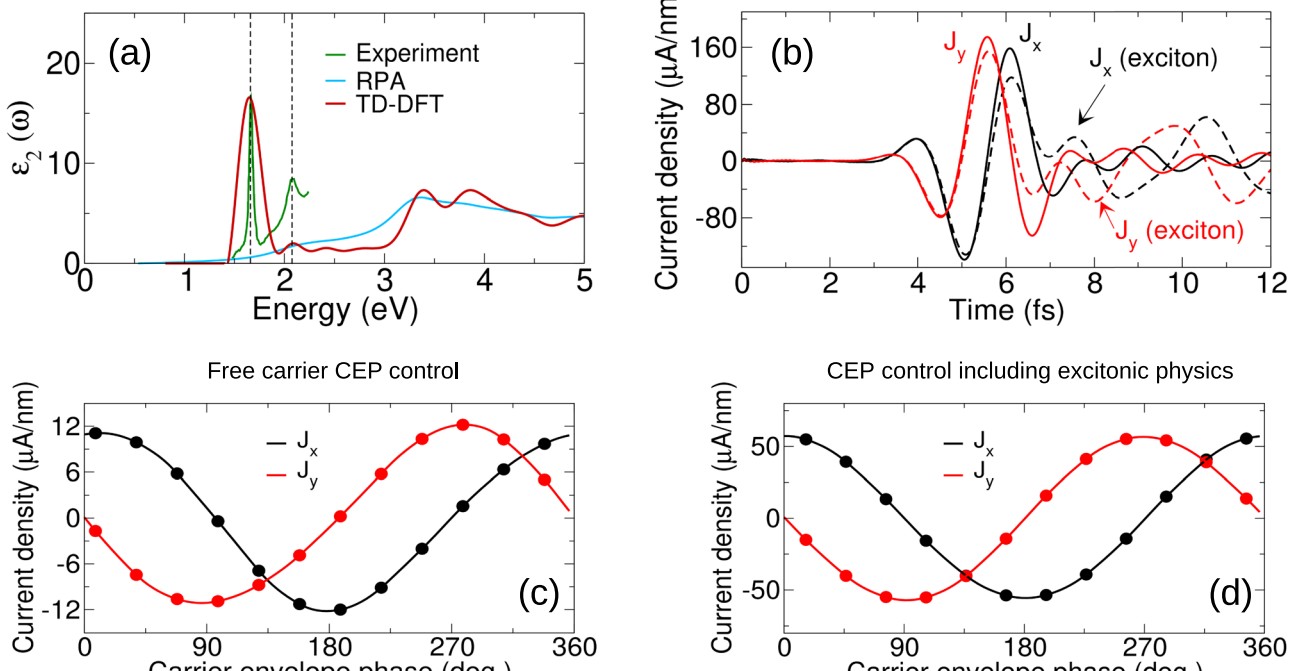

**Fig. 5 | Impact of excitonic physics on the symmetry breaking regime, example of WSe₂. a** The experimental optical absorption for WSe₂, presented alongside that calculated by time-dependent density functional theory (TD-DFT) and by the random phase approximation (RPA). The TD-DFT calculation reproduces very well the pronounced bound exciton at 1.66 eV, capturing also the position of the excitonic satellite (both indicated by vertical lines). Note that we scissors correct the local density approximation (LDA) gap of 1.44 eV to the 1.98 eV experimental gap. **b** The action of the single cycle symmetry breaking laser pulse, similar to that presented in Fig. 1(f) but with pulse frequency scaled to the 1.44 eV LDA gap, reveals an increased current response upon the inclusion of excitonic effects in the TD-DFT calculation. **c, d** Remarkably, the control over the current direction by lightform carrier envelope phase—the hallmark of the "K-pole" symmetry breaking regime of valley physics—is robust to the inclusion of excitons in the dynamics, compare Fig. 4g.

generation times of $h/\Delta$ are ultrafast: amounting to ~1 fs for gaps in the visible spectrum and ~10 fs for infrared gaps. This mechanism contrasts with that of the "hencomb" in which the components of the hybrid laser pulse address quite distinct aspects of light-matter coupling, with the necessarily sub-gap linearly polarized component precluding such generation times.

The ultrafast time scales of the "K-pole" generating pulse presented here fall between the strong field attosecond regime, far from current pump pulse technology in solids, and the multicycle regime well explored in light-matter coupling in the solid state. Experimental investigation of this domain requires phase stability over a single cycle, and such ~4 fs circularly polarized pulses already feasible in laser pulse design[32,33]. Such pulses, our work suggests, will open new vistas in the ultrafast control of matter by light.

## Methods
### Time-dependent density functional theory (TD-DFT) and the Kohn-Sham-Proca approach
Real-time TD-DFT[34,35] rigorously maps the computationally intractable problem of interacting electrons to a Kohn-Sham system of non-interacting electrons in an effective potential. The time-dependent KS equation is:

$$i\frac{\partial \psi_j(\mathbf{r},t)}{\partial t} = \left[\frac{1}{2}\left(-i\nabla - \frac{1}{c}(\mathbf{A}(t)+\mathbf{A}_{xc}(t))\right)^2 + v_s(\mathbf{r},t)\right]\psi_j(\mathbf{r},t), \quad (2)$$

where $\psi_j$ is a KS orbital and the effective KS potential $v_s(\mathbf{r},t) = v(\mathbf{r},t) + v_H(\mathbf{r},t) + v_{xc}(\mathbf{r},t)$ consists of the external potential $v$, the classical electrostatic Hartree potential $v_H$ and the exchange-correlation (XC) potential $v_{xc}$. The vector potential $\mathbf{A}(t)$ represents the applied laser field within the dipole approximation (i.e., the spatial

dependence of the vector potential is absent) and $\mathbf{A}_{xc}(t)$ the XC vector potential. This is generated by solving, in simultaneous time propagation with Eq. (2), the Proca equation

$$a_2\frac{\partial^2}{\partial t^2}\mathbf{A}_{xc}(t) + a_0\mathbf{A}_{xc}(t) = 4\pi\mathbf{J}(t), \quad (3)$$

in which the gauge invariant current is obtained from the Kohn-Sham solution at time step $t$ by

$$\mathbf{j}(\mathbf{r},t) = \text{Im}\sum_j^{occ}\psi_j(\mathbf{r},t)^*\nabla\psi_j(\mathbf{r},t) - \frac{1}{c}(\mathbf{A}(t)+\mathbf{A}_{xc}(t))\rho(\mathbf{r},t). \quad (4)$$

further details of this method are provided in ref. 21.

### Computational parameters for the TD-DFT calculations
In our calculations of WSe₂ we employ a $30 \times 30 \times 1$ k-mesh, 75 empty states corresponding to a energy cutoff of 70 eV, and the adiabatic local density approximation (LDA) as our exchange correlation functional $v_{xc}$ and the time step is 2.4 attoseconds. The electronic temperature is set to 300 K. The unit cell dimensions are $a = b = 3.31$ Å and $c = 20.0$ Å. The parameters for the Proca equation were determined to be a Proca mass of $a_0 = 0.2$ with $a_2 = 100$. For the pulse we employ a gap tuned central frequency of 1.44 eV (the LDA gap), a duration of 2.35 fs, and amplitude $A_0 = 1.37$ a.u.; these reproduce the single cycle pulse – albeit scaled to the LDA gap—shown in Fig. 1f. The pulse waveform is again that given by Eq. (8).

### An avoided crossing model ("gapped graphene")
The standard minimal model[5] capturing the important valley physics in the dichalcogenides is the nearest neighbor tight-binding model of

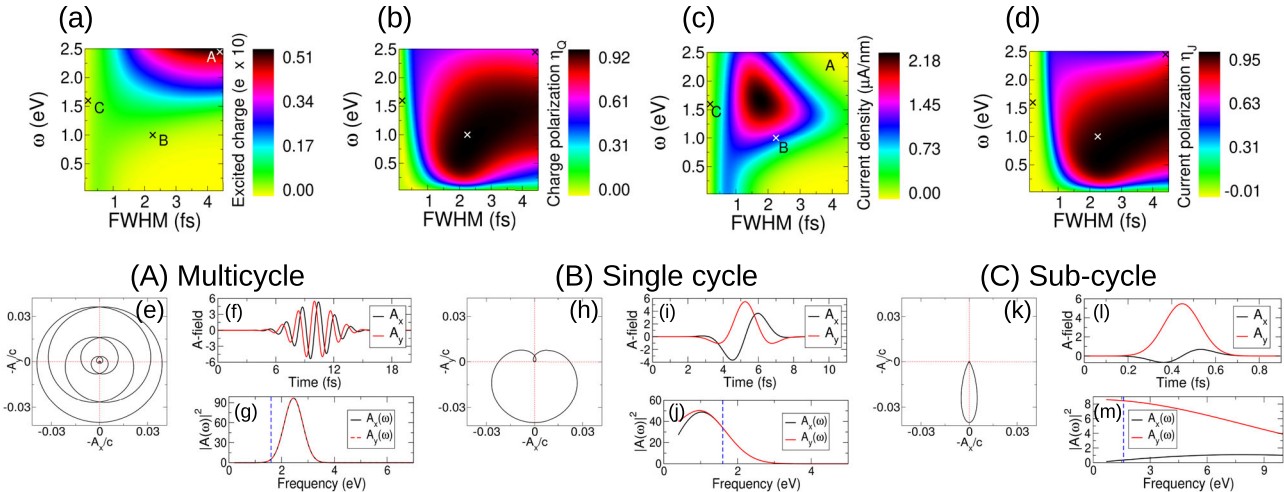

**Fig. 6 | Phase diagrams of the symmetry breaking regime of light-valley coupling.** Presented are the excited charge and its valley polarization, panels (**a**, **b**), and the excited current density and its valley polarization, panels (**c**, **d**), plotted as a function of pulse full width half maxima (FWHM) and central frequency. The emergence of valley current signals valley symmetry breaking, panel (**c**), and occurs at times corresponding to a gap tuned single cycle pulse, $h/\Delta$, ~ 1–2 fs for a gap of 1.6 eV, driven by an orientable trajectory in momentum space. At longer time multi-cycle pulses this property is lost and the valley current correspondingly vanishes.

While current persists into the sub-single-cycle attosecond regime, compare (**c**, **d**), valley polarization is lost, establishing a temporal limit on the time scale for which valley current can be generated. These distinct regimes of light-matter coupling are highlighted at the points (**a**–**c**) indicated in the phase diagrams: (A) multi-cycle, (B) single cycle, and (C) very short time sub-cycle. Panels (**e**–**m**) display for each of these cases the parametric pulse trajectory, the pulse vector potential, and its Fourier transform, revealing that each light-matter coupling regime to be associated with a distinct pulse form.

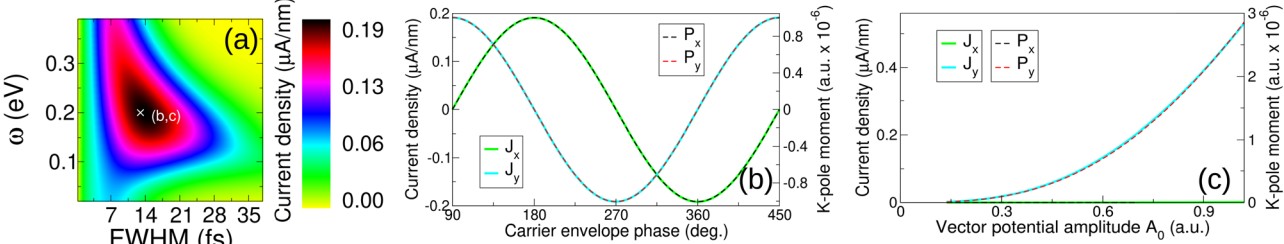

**Fig. 7 | Valley current creation in bilayer graphene.** Despite the very different low energy manifold (a so-called "Mexican hat") as compared to the avoided crossing low energy structure characteristic of transition metal dichalcogenides such as WSe$_2$, the short time emergent valley current density exhibits a very similar structure, compare panel (**a**) and Fig. 5c, highlighting the universal character of the symmetry breaking regime of circularly polarized light (the vector potential

amplitude $A_0$ = 0.685 a.u). The infrared gap (0.3 eV), however, results in a longer characteristic time scale of current generation, $h/\Delta$ ~ 14 fs. Complete control of both the valley current density direction and magnitude is again found, panels (**b**, **c**), respectively corresponding to the point indicated in (**a**), with the response driven by the "K-pole", whose moment $P$ again exactly coincides with the current.

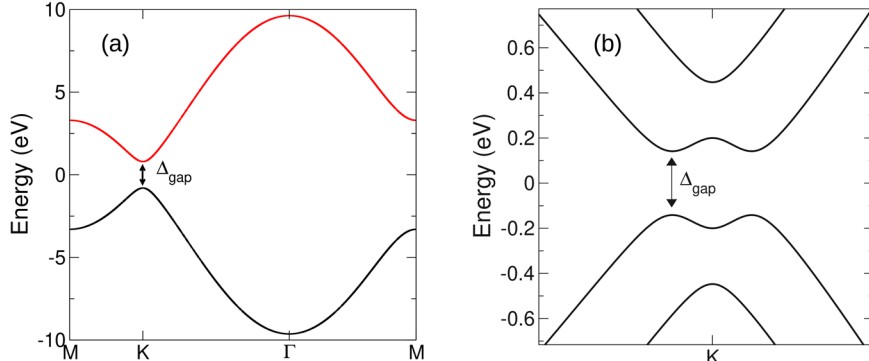

**Fig. 8 | Band structures employed in probing the short time limit of circularly polarized light pulses.** The band structure of the avoided crossing tight-binding model ("gapped graphene"), panel (**a**), and bilayer graphene, panel (**b**). In the case of bilayer graphene we present only the low energy region close to the high symmetry K point relevant for the laser induced dynamics.

graphene with a sub-lattice symmetry breaking gap $\Delta$

$$H = \begin{pmatrix} \Delta/2 & f_{\mathbf{k}} \\ f_{\mathbf{k}}^* & -\Delta/2 \end{pmatrix} \qquad (5)$$

where $f_{\mathbf{k}} = -t\sum_j e^{i\mathbf{k}.\nu_j}$ and $t = -1.4$ eV, represents the nearest neighbor hopping, with $j$ is a sum over nearest neighbor vectors $\nu_j$. The band structure for a gap of $\Delta_{gap} = 1.6$ eV, used for the data presented in the manuscript, can be found in Fig. 8a.

## Bilayer graphene

we couple two graphene layers, given by the Hamiltonian Eq. (5) but with the nearest neighbor hopping now of $t = -3.2$ eV and $\Delta = 0$ (i.e. the parameters describing graphene), by an interlayer nearest neighbor hopping of strength $t_{\perp} = 0.4$ eV:

$$H_{blg} = \begin{pmatrix} H_0 & T \\ T^\dagger & H_0 \end{pmatrix} \qquad (6)$$

where the interlayer hopping block is given by

$$T = \begin{pmatrix} 0 & t_{\perp} \\ 0 & 0 \end{pmatrix} \qquad (7)$$

The corresponding band structure can be see in Fig. 8b.

## Laser pulse

In all calculations the laser pulse has a circular polarization with Gaussian envelope centered at $t_0$:

$$\mathbf{A}(t) = A_0 \exp\left(-\frac{(t-t_0)^2}{2\sigma^2}\right)\left[\sin(\omega t + \phi_g), \cos(\omega t + \phi_g)\right] \qquad (8)$$

where $A_0$ is the pulse amplitude, $\sigma$ is related to the full width half maximum by $FWHM = 2\sqrt{2\ln 2}\sigma$, $\omega$ the central frequency of the light, and $\phi_g$ the global carrier envelope phase that determines the orientability of ultrashort circularly polarized pulses, but plays no role in standard multi-cycle circularly polarized light. Pulse parameters for individual pulses are given in the text or, in the case of the ab-initio calculations, in the computational details provided in Methods.

## Time propagation within the tight-binding Hamiltonian

The initial state is provided by a Fermi-Dirac distribution with $T = 0$, and we treat loss of quantum coherence via the simplest phenomenological model in which the density matrix, expressed in the eigenbasis at $\mathbf{k}(t)$, suffers an exponential decay of the off-diagonal density matrix elements that encode quantum interference:

$$\partial_t \rho = -i[H, \rho] + \frac{1}{T_D}(\rho - \text{Diag}[\rho]) \qquad (9)$$

where Diag[$\rho$] denotes the matrix comprising only the diagonal elements of the density matrix $\rho$, and $T_D$ is a phenomenological decoherence time which we take, following a recent experiment studying decoherence effects in graphene[36], to be 20 fs. A 300 × 300 k-mesh is employed with a time step of 20 attoseconds. The Hamiltonian $H$ in Eq. (9) is either that of the avoided crossing model, Eq. (5), or bilayer graphene, Eq. (6). For time propagation we employ the standard fourth-order Runge-Kutta method. Further details of this approach, as well as the calculation of current density, can be found in the Supplemental document.

## Data availability

All data involved in the production of the manuscript available upon reasonable request.

## Code availability

The Elk code is freely available under GNU General Public License at https://elk.sourceforge.io/.

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

## Acknowledgements

Sharma would like to thank DFG for funding through project-ID 328545488 TRR227 (projects A04), and Shallcross would like to thank DFG for funding through project-ID 522036409 SH 498/7-1. Sharma and Shallcross would like to thank the Leibniz Professorin Program (SAW P118/2021). The authors acknowledge the North-German Super-computing Alliance (HLRN) for providing HPC resources that have contributed to the research results reported in this paper.

## Author contributions

The project was designed by S.S.hal who performed the tight-binding analysis; S.S.har, D.G., and J.K.D. performed the first principles calculations using the Elk code; S.S.hal, D.G., and J.K. performed the analytical work, all authors contributed to the writing of the final manuscript.

## Funding

## Competing interests

The authors declare no competing interests.
