## [Peer Review File · Nature Communications]

REVIEWER COMMENTS

Reviewer #1 (Remarks to the Author):

Sharma and Shallcross theoretically predict the excitation of valley-polarized currents from circularly polarized, few cycle, femtosecond pulses. The authors attribute the emergence of macroscopic incoherent intraband currents to the Berry curvature in connection with a light-induced symmetry breaking. Computed results for gapped and bilayer graphene are presented. The authors also quantify the efficiency of current generation as a function of the pulse parameters and find the optimum to be achieved for resonant excitation of the gap with a circularly-polarized, few-cycle pulse.

The predicted ultrafast generation of valley-polarized currents would be of interest to the community and experimental verification of this prediction could open new intriguing possibilities for lightwave electronics. The authors identify optimum excitation conditions to generate currents which could be tested experimentally. However, in my opinion, the paper requires significant revisions before publication since some key questions to judge the validity of several statements are not discussed properly.

A) The central argument of the paper is that a macroscopic current is being excited due to a difference in the "recorded" Berry curvature in the few-cycle limit compared to multi-cycle pulses. I find the explanation for this observation provided in lines 95-115, and in particular lines 98-101, insufficient and the arguments not comprehensible. The following key questions should be discussed much more clearly:

- What do the authors mean with "recorded $\Omega(k)$ "?
- Why is it independent of $k(0)$ for multi-cycle pulses and not for few-cycle pulses?
- How does this affect the final $|c(k)|^2$ distribution which shows the symmetry breaking?

To me, these seem to be the key questions that the paper tries to address, and should be addressing, to explain the observed effect. Without these arguments it is not conclusively demonstrated that the currents are indeed created by the Berry connection.

B) The authors discuss that the creation of currents works best for excitations at the gap. In realistic materials, however, the optical response near the gap is dominated by excitonic

states which also significantly alter electronic occupations ($|c(k)|^2$). As far as I understand, the model used by the authors does not take into account Coulomb interactions or excitonic effects. How does the Coulomb interaction affect the excited currents? This is an important piece of information to determine if experimental verification of the prediction is viable.

Beyond the two comments above which question the main conclusions of the paper, there also seem to be some inaccuracies in the paper that I suggest to address before publication:

1) The authors quote explicit values for the excited currents. However, I would expect this value to be an extensive quantity which depends on the geometry of the device measuring the current. How exactly is the value of the current computed and what assumptions are made for it? In addition, in line 177 the authors state to have predicted "massive ... valley currents". This statement is not backed by the manuscript because the quoted value of 2.6 μA is not put into context or compared to other approaches. This is in addition to the unclear definition of the value as described above.

2) In line 84 the authors quote that " $D(k)=0$ " for the multi-cycle pulse. This is probably not true. Since the pulse still has a finite pulse length, I would expect a small amount of symmetry breaking to occur which prevents $D(k)$ to vanish exactly. It might be instructive to add a figure of the maximum of $|D(k)|$ as function of the FWHM. This information is in principle contained in Fig.4 but might help the discussion of Figs 1 and 2.

3) In line 55 the authors state that they "include realistic decoherence" in their model. At the same time, it is described as "the simplest phenomenological model" in the Methods section. The authors should describe it as such since a truly realistic dephasing should include, for example, excitation-induced dephasing which is clearly not the case here. Otherwise, some discussion is needed to clarify why a phenomenological dephasing is "realistic" here.

4) In line 56 the authors refer to the Methods section for details on the computational model. However, the description provided in the Methods section seems inadequate to reproduce the results. A more elaborate discussion in the supplement or at least a citation is needed here. For example, what is and what is not included in the Hamiltonian H in Eq. (4)? Also, it would be good practice to specify the parameters of the vector potential (line 198) as well as the temperature that is assumed for the initial state (line 200). This would be helpful to ensure the results are reproducible by other groups.

5) In line 187 the authors seem to suggest that few-cycle pulses are not available to date. This is outside my field of expertise, however, I was under the impression that several labs would be able to generate such pulses nowadays. At the very least, citations should be added to show the state-of-the-art in this respect.

Reviewer #2 (Remarks to the Author):

In this manuscript, the authors propose lightwave manipulation of the valley current in valleytronics materials, such as gapped graphene and biased bilayer graphene. The coupling between circularly polarized light and valley has been well studied in previous works. However, since both the valleys in “gapped graphene” and the circularly polarized light have the C_{3z} rotation, the light while can generate significant valley charge, it can not produce net valley current, as the current is forbidden by C_{3z} .

In this work, the authors show that at femtosecond time scale, this well-established wisdom no longer applies. This is demonstrated by studying the valley charge and valley current excited by long-time multicycle (23.5 fs duration) and ultrashort single cycle (2.3 fs) pulses. The authors find that the net valley current vanishes for a long time multicycle but appears for an ultrashort single cycle. Then, the authors discuss the underlying physics and show the finite valley current results from the fact that the vector potential A of light is a vector and its trajectory in a single cycle breaks the C_{3z} symmetry. But, the average of the A -trajectories over multiple periods would respect the C_{3z} .

I think this is a nice work that brings insights into the physics in the ultrafast limit of light-matter coupling, and would attract broad interest in the fields of both optic physics and condensed matter physics. Before recommending its publication, I have some comments that should be addressed by the authors.

First of all, the authors discussed that the valley current is from intra-band contribution.

As we know, under circularly polarized light, both valence and conduction bands have free charges. Hence, does the intra-band contribution refer to the valence or conduction band? I hope the authors clarify this. Besides, what about the inter-band contribution?

Second, in line 150 of this manuscript, the author mentions "while at shorter times reduction in the weight of gap tuned frequency components." Could you please further explain the meaning of the gap tuned frequency components? And why are they special?

Third, in my understanding, the ultrafast dynamic refers to the light with high frequency. Hence, how does such ultrafast light generate valley charge and then the valley current in the biased bilayer graphene, as the band gap of the biased bilayer graphene is rather small?

Reviewer #3 (Remarks to the Author):

This paper studies the coupling of circularly polarized light to valley current in two dimensional semiconductors. They show that in the few cycle limit of light, the direction and magnitude of valley current can be controlled by the orientation and magnitude of light. They find that the effect exists both in visible spectrum gaps and infrared gaps.

The idea of creating valley current by light seems interesting and adds new results in the whole valleytronic panorama. The authors support their idea by introducing the concept of momentum space valley dipole and using the density matrix technique.

However, prior to publications, the authors should deal with some modifications of their current manuscript. I address here below my requests:

1) In Reference 14, the authors have proposed THz induced giant spin and valley current. What are the differences of results between this manuscript and reference 14? The authors should address the concern both in the reply and revised manuscript.

2) The expression of density $D(k)$ from Eq. (1) is confusing. It seems that there are only two terms in the summation. How does it preserve C_3 symmetry? The authors should provide a schematic plot to clarify the expression.

3) The authors discuss the long time multi cycle and ultrashort single cycle limit in Figure 2, respectively. What about the middle regime? The authors should provide data to show the evolution of physical properties from ultrashort to long time cycle limit.

4) In Figure 2 (d)-(f), C_3 symmetry is broken. Is there any intuitive physical understanding of the symmetry breaking?

5) In Figure 3, authors mention that the orientation can be controlled by the pulse carrier envelope phase (CEP). How to tune CEP experimentally? The authors should provide more discussions on the experimental realization of the effect in the manuscript.

6) In line 149-151, there is a statement "at longer times the multi cycle pulses lack orientability, while at shorter times reduction in the weight of gap tuned frequency components, as well as loss of curvature of the pulse trajectory in momentum space". The authors should provide more details to explain the statement.

7) According to Figure 1, it seems that valley current is accompanied by valley charge. Is it possible to realize the valley current without valley charge, that is, independent control of valley charge and valley current?

8) In the manuscript, circularly polarized light is considered. What will happen if we apply the linearly polarized light?

9) In the section of Methods, there is a system named gapped graphene. However, for monolayer gapped graphene, the gap value can not be as large as 1.6 eV. The authors should change the name.

10) In the studies of light-induced valley charge, excitons and trions are formed as a result of Coulomb interaction. How does such interaction affect the physics proposed in the manuscript?

Reviewer #1 (Remarks to the Author)

Sharma and Shallcross theoretically predict the excitation of valley-polarized currents from circularly polarized, few cycle, femtosecond pulses. The authors attribute the emergence of macroscopic incoherent intraband currents to the Berry curvature in connection with a light-induced symmetry breaking. Computed results for gapped and bilayer graphene are presented. The authors also quantify the efficiency of current generation as a function of the pulse parameters and find the optimum to be achieved for resonant excitation of the gap with a circularly-polarized, few-cycle pulse.

The predicted ultrafast generation of valley-polarized currents would be of interest to the community and experimental verification of this prediction could open new intriguing possibilities for lightwave electronics. The authors identify optimum excitation conditions to generate currents which could be tested experimentally. However, in my opinion, the paper requires significant revisions before publication since some key questions to judge the validity of several statements are not discussed properly.

We thank the Referee for the extremely careful reading of our paper, and for the many useful suggestions. In particular, in response to the comments of the Referee, we have now:

(i) Rewritten and hopefully clarified the section of our manuscript detailing the relation between the few cycle pulse and the Berry curvature, and how this leads to an emergent regime characterized by the ultrafast generation of pure valley current.

(ii) Performed time-dependent density functional theory (TD-DFT) calculations of the laser induced dynamics treating excitons and free carriers on an equally footing. These calculations for WSe_2 reveal the same remarkable emergent carrier envelope phase control of the symmetry breaking current that we found employing minimal tight-binding models in which (as the referee correctly points out) excitonic effects were not included.

We appreciate the Referee drawing our attention to these points, and believe in answering them the paper has been greatly improved.

A) The central argument of the paper is that a macroscopic current is being excited due to a difference in the "recorded" Berry curvature in the few-cycle limit compared to multi-cycle pulses. I find the explanation for this observation provided in lines 95-115, and in particular lines 98-101, insufficient and the arguments not comprehensible. The following key questions should be discussed much more clearly:

- What do the authors mean with "recorded $\Omega(k)$ "?
- Why is it independent of $k(0)$ for multi-cycle pulses and not for few-cycle pulses?
- How does this affect the final $|c(k)|^2$ distribution which shows the symmetry breaking?

To me, these seem to be the key questions that the paper tries to address, and should be addressing, to explain the observed effect. Without these arguments it is not conclusively demonstrated that the currents are indeed created by the Berry connection.

We agree with the Referee that this was not clearly explained, and have now rewritten this part of the paper. Lines 104-131 of the paper have been substantially rewritten, and a new figure incorporated into this section to illustrate that the origin of the few cycle symmetry

breaking resides in the “shape” of the dynamical evolution of crystal momenta and the underlying valley Berry curvature.

We now address both the three specific questions of the Referee, and the more general point of demonstrating that the currents are created by the Berry curvature.

(i) By the “recorded $\Omega(\mathbf{k})$ ” we simply mean the Berry curvature $\Omega(\mathbf{k}(t))$ at each point in time during the light induced evolution of momentum from an initial momenta according to the Bloch acceleration theorem $\mathbf{k}(t) = \mathbf{k}(0) - \mathbf{A}(t)/c$.

Fig. 1: *The physical mechanism of emergent short time symmetry breaking.* (a) For long time pulses, the laser driven $\mathbf{k}(t)$ -trajectories of two initial momenta $(0, +k_y)$ and $(0, -k_y)$, indicated by the crosses in panel (a), approximately map into each other under $k_y \rightarrow -k_y$ and, as a result, the Berry curvature “recorded” along the dynamical trajectories exhibit similar but phase shifted oscillations, panel (b), generating ultimately nearly identical final conduction band occupations $|c_{\mathbf{k}}|^2$, panel (c). Reduction in pulse duration to ~ 1 -2 fs drives light-matter coupling to a symmetry breaking regime, (d-f), in which the few cycle laser driven trajectories for initial momenta $(0, \pm k_y)$ now lack $k_y \rightarrow -k_y$ reflection symmetry, panel (d), with, as a consequence, very different Berry curvatures “recorded” along the two dynamical trajectories, panel (e). The $(0, +k_y)$ initial momenta – whose dynamical $\mathbf{k}(t)$ trajectory drives it *into* the region of high Berry curvature near the K point – has increased conduction band charge $|c_{\mathbf{k}}|^2$ as compared to the $(0, -k_y)$ initial momenta whose dynamical trajectory drives it *away* from the region of high Berry curvature, panel (f). The laser induced charge excitation inherits this $k_y \rightarrow -k_y$ symmetry breaking, leading to the finite “K-pole” structure exhibited by momentum resolved excited state population shown in Fig. 2(c,d) of this reply.

(ii,iii) For multicycle pulses the trajectory $\mathbf{k}(t)$ induced by light describes circular motion in momentum space. For two initial momenta $(0, \pm k_y)$, indicated by crosses in Fig. 1a of this reply (Fig. 3 of the manuscript), these trajectories approximately map into each other under $k_y \rightarrow -k_y$ and, as the Berry curvature also respects $k_y \rightarrow -k_y$, the curvature “recorded” by each trajectory $\Omega(\mathbf{k}(t))$ will thus be very similar. This is shown in Fig. 1b in which the Berry curvature $\Omega(\mathbf{k}(t))$ is plotted as a function of time for each initial momenta. The oscillation results from the two initial momenta $(0, \pm k_y)$ successively evolving closer to the region of high Berry curvature at K and subsequently away from it, as a result of the circular motion entailed by $\mathbf{k}(t)$, while the $\pi/2$ phase shift has its origin in the fact that while the $(0, +k_y)$ trajectory evolves towards K the $(0, -k_y)$ trajectory necessarily evolves away from K. Regions of high Berry curvature are associated with excitation of charge to the conduction band, however over many cycles the effect for the two initial momenta $(0, \pm k_y)$ averages out and, as a result, and as may be seen in Fig. 1c, the conduction band occupations post laser pulse are nearly identical, $|c_{+k_y}|^2 \sim |c_{-k_y}|^2$.

For the single cycle pulse the situation is very different. Now the $\mathbf{k}(t)$ trajectory executes essentially only one loop, which may either be *into* the region of high Berry curvature, or *away* from it, see Fig. 1d. For the $(0, +k_y)$ initial momenta, light evolution of momenta into high $\Omega(\mathbf{k})$ will result in increased excitation to the conduction band as compared to the $(0, -k_y)$ initial condition, which evolves away from the region of high Berry curvature. This can be seen in Fig. 1e,f. We now have $|c_{+k_y}|^2 > |c_{-k_y}|^2$ and thus the near $k_y \rightarrow -k_y$ equality of the conduction band occupations found for the multi-cycle case is broken.

As this will occur for every such $(0, \pm k_y)$ pair, the laser induced excited charge will inherit this symmetry breaking, exactly as recorded by the symmetry breaking density $D(\mathbf{k})$ for the few cycle pulse, see Fig. 2(c,d) of both reply and manuscript, but found to be absent for the multicycle pulse, Fig. 2(a,b), in which the excitation preserves the underlying valley C_3 symmetry, and $D(\mathbf{k})$ is 10^{-9} smaller in magnitude.

Fig. 2: “K-pole” dipole moment characterizing the emergence of an early time symmetry breaking regime of valley coupling by circularly polarized light. (a-b) For long time pulses C_3 valley symmetry is preserved in the laser induced charge excitation and the symmetry breaking density $D(\mathbf{k})$, is therefore negligible being $< 10^{-8}$. At pulse durations of ~ 1 -2 fs light-matter coupling is driven to a symmetry breaking regime, (c-d), with the laser induced charge excitation exhibiting a finite “K-pole” structure to the momentum resolved excited state population with clear breaking $+k_y \rightarrow -k_y$ mirror symmetry. In this regime the lightform generates a valley current.

We hope this now makes clear the connection between the Berry curvature and the dynamics in the few and multi-cycle pulses. We have incorporated these changes into the paper, which we hope is now significantly clearer in respect of explaining the underlying mechanism.

To further confirm this relation between few cycle trajectory and the Berry curvature we have performed calculations for a series of amplitudes of such few cycle pulses, see Fig. 3

of this reply below. Increasing the vector potential amplitude of the few cycle pulse leads to a monotonic increase in valley current, Fig. 3a in which three values are highlighted as points (A-C). This increase can be directly correlated with the “shape” of the light induced evolution of $\mathbf{k}(t)$, Fig. 3b, in which we present the $\mathbf{k}(t)$ trajectory for the 3 points highlighted in Fig. 3a. As the single dynamical loop executed by the light induced dynamics increases in size, impinging further into the region of high Berry curvature close to the K point, the conduction band excitation increases accordingly; Fig. 3c shows the conduction band occupation for each of these amplitudes, while Fig. 3d presents the Berry curvature recorded along each trajectory $\Omega(\mathbf{k}(t))$. At the same time, “fat trajectory” plots – Fig. 3(e-g) in which the width of the dynamical trajectory corresponds to the rate of change in conduction band excitation – make clear that the increase in conduction band population precisely corresponds to the those portions of the dynamical evolution in which $\mathbf{k}(t)$ is associated with regions of high Berry curvature.

Fig. 3: (a) The symmetry breaking valley current induced by the single cycle circularly polarized pulse presented in Fig. 1(f-j) of the manuscript, shown as a function of the vector potential amplitude A_0 . The current is seen to increase with the amplitude of the ultrafast pulse waveform, and for three representative values, denoted A-C in panel (a), we present in panel (b) the parametric plot of the dynamical evolution of the crystal momentum, $\mathbf{k}(t)$, induced by the laser pulse for an initial \mathbf{k} -vector $(0, +k_y)$. The correspond time dependence of the conduction band occupation and the (normalized) Berry curvature at $\mathbf{k}(t)$ are presented in panels (c) and (d), respectively, revealing that the greater the Berry curvature “recorded” along the trajectory, $\Omega(\mathbf{k}(t))$, the greater the corresponding conduction band occupation. Panels (e-g) display “fat trajectory” plots for each of the amplitudes A-C indicated in (a), in which the width of the trajectory lines indicates the rate of change of the occupation of the conduction band.

B) The authors discuss that the creation of currents works best for excitations at the gap. In realistic materials, however, the optical response near the gap is dominated by excitonic states which also significantly alter electronic occupations ($|c(\mathbf{k})|^2$). As far as I understand, the model used by the authors does not take into account Coulomb interactions or excitonic effects. How does the Coulomb interaction affect the excited currents? This is an important piece of information to determine if experimental verification of the prediction is viable.

The Referee is completely correct in that in treating the dynamics via minimal tight-binding models evolving according to the von Neumann equation, Eq. 6 of the methods, we do not include excitonic effects. To remedy this we have performed time dependent density functional theory (TD-DFT) calculations of WSe_2 employing a recently proposed Kohn-Sham-Proca scheme [arXiv:2401.16140], shown to capture both the bound excitonic peaks and spectral weight renormalization of the linear response regime, but more importantly, key features of non-equilibrium excitonic physics such as laser pump induced “bleaching” (reduction in absorption) of the excitonic peak and the appearance of excitonic side bands. This methodology, we note, treats excitons and free carriers on an equal footing, making it the ideal platform for investigation of the impact of excitons on early time pump laser dynamics.

Our findings are shown in Fig. 4 of this reply (and a new Fig. 5 of the manuscript along with associated addition to the text, lines 162-184). The linear response absorption spectrum of WSe_2 , Fig. 4a, is seen to be very well captured: the excitonic peak at 1.66 eV is reproduced in excellent agreement with experiment, but also the position (though not amplitude) of a second satellite peak. Application of a single cycle pump laser pulse generates a pronounced current response both for the TD-DFT simulation including excitonic effects as well as for the “vanilla” TD-DFT simulation excluding excitonic effects, Fig. 4b. Interestingly, the current response is greater in magnitude with the inclusion of excitonic physics; as shown in the Supplemental via the time-dependent density of states this occurs as the strong field pulse resulting in ionization of excitons leading to an increase in the free carrier population, changing the value of the conduction band $|c_k|^2$ as suggested by the Referee. This increase in free carrier population, also noticed in a recent model investigation [arXiv:2307.14693], indicates an intrinsic coupling of exciton and free carrier populations under pump laser conditions.

Fig. 4: Impact of excitonic physics on the symmetry breaking regime, example of WSe_2 : The experimental optical absorption for WSe_2 , presented alongside that calculated via time-dependent density functional theory (TD-DFT) and by the random phase approximation (RPA). The TD-DFT calculation reproduces very well the pronounced bound exciton at 1.66 eV, capturing also the position of the excitonic satellite (both indicated by vertical lines). Note that we scissor correct the local density approximation (LDA) gap of 1.44 eV to the 1.98 eV experimental gap. (b) The action of a single cycle symmetry breaking laser pulse, similar to that presented in Fig. 1(f) of the manuscript but with the frequency scaled to the 1.44 eV LDA gap, reveals an increased current response upon the inclusion of excitonic effects in the TD-DFT calculation. Remarkably, the control over the current direction by carrier envelope phase - the hall mark of the “K-pole” symmetry breaking regime of valley physics -- is robust to the inclusion of excitonic effects in the dynamics.

The critical test is whether this current is CEP controllable in the TD-DFT simulations – this is the hallmark of the orientable trajectory in momentum space and thus of the symmetry breaking regime. As can be seen in Fig. 4(c,d), this is clearly so: the sinusoidal oscillation found in the minimal tight-binding avoided crossing model ("gapped graphene") is reproduced by the TD-DFT current both including excitonic effects and in their absence.

It would thus appear that the excitonic fraction at short times thus follows the same "K-pole" symmetry breaking as the free carriers acting, as may be seen from comparison of Fig. 4(c,d), to enhance the associated valley current. We thus conclude that the effect we report is robust both to (i) realistic simulation in a specific valley active material as well (ii) to the inclusion of excitonic effects.

We thank the Referee for pointing out the importance of including excitonic effects in our analysis: we believe that by demonstrating that an *ab-initio* calculation including excitonic physics also exhibits key characteristics of the ultrafast symmetry breaking regime of light-valley coupling physics considerably strengthens the case our paper makes.

Beyond the two comments above which question the main conclusions of the paper, there also seem to be some inaccuracies in the paper that I suggest to address before publication:

1) The authors quote explicit values for the excited currents. However, I would expect this value to be an extensive quantity which depends on the geometry of the device measuring the current. How exactly is the value of the current computed and what assumptions are made for it? In addition, in line 177 the authors state to have predicted "massive ... valley currents". This statement is not backed by the manuscript because the quoted value of 2.6 μA is not put into context or compared to other approaches. This is in addition to the unclear definition of the value as described above.

We agree with the Referee and have replaced all usage of "current" by "current density". In the previous version we multiplied the latter by a characteristic length of the order of the lattice constant to obtain an "feel" of the corresponding currents in a nanodevice context, however, we agree with the Referee that it is better to present directly the current densities.

The statement of "massive" valley currents was motivated by comparison of our results for bilayer graphene to experiments on graphene that, utilizing different and longer time waveforms, typically report currents on the few nA to pA scale (see e.g. doi:10.1038/nature23900 and *Nano Lett.* 2021, 21, 9403–9409), much smaller than the currents implied by the current densities we report here. However, we now have removed this description as the comparison is not a direct one: experiments measure diffusive longer time currents (that, nevertheless, inherit key properties of the early time coherent current).

2) In line 84 the authors quote that " $D(k)=0$ " for the multi-cycle pulse. This is probably not true. Since the pulse still has a finite pulse length, I would expect a small amount of symmetry breaking to occur which prevents $D(k)$ to vanish exactly. It might be instructive to add a figure of the maximum of $|D(k)|$ as function of the FWHM. This information is in principle contained in Fig.4 but might help the discussion of Figs 1 and 2.

The Referee is correct that the symmetry breaking density $D(\mathbf{k})$ will – for a pulse of finite duration – never be exactly zero, and in Fig. 2 of the manuscript (reproduced above as Fig. 2 of the reply) we now explicitly show the $D(\mathbf{k})$ function for the multi-cycle pulse with its own scale, revealing it to be of the order of 10^{-9} smaller than for the symmetry breaking single cycle pulse. We thank the Referee for pointing this out.

Fig. 5: Dependence of (a) the current and (b) the maximum value of the absolute value of $D(\mathbf{k})$, the symmetry breaking density, for times that interpolate between the long time multi-cycle high symmetry regime and the short time few cycle symmetry breaking regime. The current is seen to fall continuously to zero, behaviour in close correspondence with $\max |D(\mathbf{k})|$, indicating that this symmetry breaking density is correlated to the short time emergence of a valley current generated by circularly polarized light.

We also agree with the Referee that, although it is implicit in the phase diagrams we present, it is useful to also present a figure of the maximum of $|D(\mathbf{k})|$ as a function of pulse duration. We present this above as Fig. 5 of the reply, in which we interpolate between the single cycle pulse (full width half maxima i.e. duration ~ 2 fs) and the long time pulse (duration > 10 fs). Panel (a) of this figure displays the current characteristic of the symmetry breaking regime that emerges at short times, while in panel (b) we present the maximum value of $|D(\mathbf{k})|$. The close correspondence between these two again indicates the key role that the symmetry breaking density $D(\mathbf{k})$ plays in the ultrafast limit of light-valley coupling physics. This figure we have now included in the Supplemental as a complementary point of view to the implicit presentation of this information via the phase diagrams, and we thank the Referee for suggesting this addition.

3) In line 55 the authors state that they "include realistic decoherence" in their model. At the same time, it is described as "the simplest phenomenological model" in the Methods section. The authors should describe it as such since a truly realistic dephasing should include, for example, excitation-induced dephasing which is clearly not the case here. Otherwise, some discussion is needed to clarify why a phenomenological dephasing is "realistic" here.

We have now made the description of what is indeed a phenomenological model consistent throughout the manuscript. It is realistic only in the sense that the effective decoherence time we use for graphene type systems – 20 fs – represents a value obtained in a recent experiment, now cited as Ref. 32 of the revised manuscript; we have now a statement to indicate this fact (lines 287-288). We thank the Referee for catching this inconsistency of our presentation.

4) In line 56 the authors refer to the Methods section for details on the computational model. However, the description provided in the Methods section seems inadequate to reproduce the results. A more elaborate discussion in the supplement or at least a citation is needed here. For example, what is and what is not included in the Hamiltonian H in Eq. (4)? Also, it would be good practice to specify the parameters of the vector potential (line

198) as well as the temperature that is assumed for the initial state (line 200). This would be helpful to ensure the results are reproducible by other groups.

We agree with the Referee and have restructured and considerably expanded the Methods section, now particularly important as we include *ab-initio* calculations in the analysis of the short time symmetry breaking regime of circularly polarized light. The Hamiltonian H in the von Neumann equation, Eq. 6 of the methods, has been identified (lines 289-290), and the description of the pulse waveform has been improved (lines 275-281), and at all places in the manuscript laser pulse information has been given in full that should allow independent reproduction of our calculations. We employed a zero temperature initial state for the tight-binding calculations, and 300 K for the *ab-initio* calculations, and this has also now been explicitly stated in Methods. Finally, we refer in methods to the Supplemental document for a more in-depth description of our time propagation of the von Neumann equation and the current density formalism (Sections 1 and 5). We thank the Referee for drawing our attention to the brevity of our Methodological description, and hope that it now offers the complete set of data relevant for reproducing our calculations.

5) In line 187 the authors seem to suggest that few-cycle pulses are not available to date. This is outside my field of expertise, however, I was under the impression that several labs would be able to generate such pulses nowadays. At the very least, citations should be added to show the state-of-the-art in this respect.

The Referee is correct, as we now realize after discussion with a experimental colleague who can make these very pulses. We have added citations, Ref. 28 and 29 of the revised paper, to indicate that few femtosecond pulses are available in the visible range that have full control over the carrier envelope phase (lines 247-249).

Reviewer #2 (Remarks to the Author):

In this manuscript, the authors propose lightwave manipulation of the valley current in valleytronics materials, such as gapped graphene and biased bilayer graphene. The coupling between circularly polarized light and valley has been well studied in previous works. However, since both the valleys in “gapped graphene” and the circularly polarized light have the C_{3z} rotation, the light while can generate significant valley charge, it can not produce net valley current, as the current is forbidden by C_{3z} .

In this work, the authors show that at femtosecond time scale, this well-established wisdom no longer applies. This is demonstrated by studying the valley charge and valley current excited by long-time multicycle (23.5 fs duration) and ultrashort single cycle (2.3 fs) pulses. The authors find that the net valley current vanishes for a long time multicycle but appears for an ultrashort single cycle. Then, the authors discuss the underlying physics and show the finite valley current results from the fact that the vector potential A of light is a vector and its trajectory in a single cycle breaks the C_{3z} symmetry. But, the average of the A -trajectories over multiple periods would respect the C_{3z} .

I think this is a nice work that brings insights into the physics in the ultrafast limit of light-matter coupling, and would attract broad interest in the fields of both optic physics and condensed matter physics. Before recommending its publication, I have some comments that should be addressed by the authors.

We thank the Referee for the careful reading of the manuscript and the several suggestions towards its improvement that have been made.

We have acted on these, in particular by:

1. Clarifying the discussion around the phase diagrams (Fig. 4 of the original manuscript, Fig. 6 of the revised manuscript, and reproduced below as Fig. 7 of this reply).
2. By expanding our discussion of inter-band and intra-band currents in the manuscript and its supplemental. We now show the the relation of carrier envelope phase ϕ_g to the direction θ_J of the emergent valley current, $\theta_J = \phi_g$, holds both for the intra-band current and the total (with a time dependent shift).

First of all, the authors discussed that the valley current is from intra-band contribution. As we know, under circularly polarized light, both valence and conduction bands have free charges. Hence, does the intra-band contribution refer to the valence or conduction band? I hope the authors clarify this. Besides, what about the inter-band contribution?

The current density is obtained from the momentum gradient of the Hamiltonian as

$$\mathbf{j}(t) = \frac{1}{V_{UC}} \sum_{\mathbf{q}} \langle \Psi_{\mathbf{q}}(t) | \nabla_{\mathbf{p}} H(\mathbf{p}, t) | \Psi_{\mathbf{q}}(t) \rangle \quad (\text{Equation 1})$$

where \mathbf{q} runs over vectors in the 1st Brillouin zone of area V_{UC} and $|\Psi_{\mathbf{q}}(t)\rangle$ is the time dependent evolving system ket at \mathbf{q} . We can expand in a basis of eigenstates $|\Phi_{i\mathbf{k}(t)}\rangle$ (corresponding eigenvalues $\epsilon_{i\mathbf{k}(t)}$) at the time dependent crystal momentum $\mathbf{k}(t)$ given by the Bloch acceleration theorem $\mathbf{k}(t) = \mathbf{q} - \mathbf{A}(t)/c$.

Inserting this into Eq. 1 above then yields separate intra- and inter-band component

$$\mathbf{j}_{intra}(t) = \frac{1}{V_{UC}} \sum_{i\mathbf{q}} |c_{i\mathbf{k}(t)}|^2 \nabla_{\mathbf{p}} \epsilon_{i\mathbf{k}(t)} \quad (\text{Equation 2})$$

Fig. 6: The vector potential $\mathbf{A}(t)$ of a near single cycle circularly polarized laser pulse is presented in panel (a), with the corresponding light induced valley resolved current densities (denoted by subscripts K and K^*) shown in panel (b). Shown are both the total current as well as its *intra*band component, as indicated in the caption. Note that while these differ – the total exceeds the intraband due to the presence of the interband component – the same physics is seen: nearly complete valley polarization of the current generated on the same femtosecond time scale with $|\mathbf{j}_K| \gg |\mathbf{j}_{K^*}|$.

Fig. 7: Carrier envelope phase control over the total current. (a) The total current at 11.5 fs, indicated by the vertical line in Fig. 6, is shown as a function of carrier envelope phase with all other pulse parameters held fixed. Evidently complete control over total current direction can be obtained via the carrier envelope phase. Except for a constant shift this is exactly the form found for the intraband component, panel (b).

and

$$\mathbf{j}_{inter}(t) = -\frac{1}{V_{UC}} \sum_{ij\mathbf{q}} c_{i\mathbf{k}(t)}^* c_{j\mathbf{k}(t)} \epsilon_{j\mathbf{k}(t)} \langle \nabla_{\mathbf{p}} \Phi_{i\mathbf{k}(t)} | \Phi_{j\mathbf{k}(t)} \rangle + c.c. \quad (\text{Equation 3})$$

where $c_{i\mathbf{k}(t)}$ are the expansion coefficients. In Eq. 2 the sum indeed runs over both valence and conduction band as the Referee suggests. In fact, for the 2-band model we consider these are complementary: the conduction and valence band occupations sum up to unity and have opposite band velocities so they essentially both contribute equally to the current.

The question as to the role of the inter-band current is important, and we have addressed this now more carefully in the Supplemental, Sections 1 and 2. As can be seen from Eq. 3 this current term is considerably more complex, however it plays a role only at early times: the presence of the \mathbf{k} -vector dependent phase terms in Eq. 3 – as opposed to simply the occupations, as in Eq. 2 for the intraband current – renders the interband component susceptible to destructive interference. Indeed, this is what we typically see, as illustrated above in Fig.6 of this reply in which the total current (including interband) is seen to limit to its intraband component as interference effects take hold at longer times.

We have also verified that at early times in which the total and intraband current do differ, the key physics of the symmetry breaking regime we identify in the manuscript is carried by both components of the valley current. This we show in Fig. 7 of this reply above, now Fig. 2 of the Supplemental, in which the CEP control of the emergent valley current – a manifestation of the orientability of the trajectory of the light induced momentum evolution – is shown to hold not only for the intraband component, but also for the total i.e. intraband and interband. The origin of the shift between the intraband and interband current density sinusoidal CEP dependence, that can be seen by comparing Fig. 7a and Fig. 7b, we provide further analysis of in the Supplemental, Section 2.

We thank the Referee for pointing out that our description of the current could be expanded upon, and in doing so we feel the paper has been improved. In the section on control of the emergent valley current we have expanded the text (lines 156-160) to point out that while we present the intraband current (i) the full current shows identical features and (ii) the full current limits to the intraband component after 10s of femtoseconds.

Fig. 8: Phase diagrams of the symmetry breaking regime of light-valley coupling. Presented are the excited charge and its valley polarization, panels (a,b), and the excited current and its valley polarization, panels (c,d), plotted as a function of pulse full width half maxima (FWHM) and central frequency. The emergence of valley current signals valley symmetry breaking, panel (c), and occurs at times corresponding to a gap tuned single cycle pulse, \hbar/Δ , ~ 1 -2 fs for a gap of 1.6 eV, driven by an orientable trajectory in momentum space. At longer time multi-cycle pulses this property is lost and the valley current correspondingly vanishes. While current persists into the sub-single-cycle attosecond regime, compare (c,d), valley polarization is lost, establishing a temporal limit on the time scale for which valley current can be generated. These distinct regimes of light-matter coupling are highlighted at the points indicated in the phase diagrams: multi-cycle, single cycle, and very short time sub-cycle. Panels (e-m) display for each of these cases the parametric $\mathbf{A}(t)$ -trajectory, the pulse vector potential, and its Fourier transform, revealing that each light-matter coupling regime to be associated with a distinct pulse topology.

Second, in line 150 of this manuscript, the author mentions "while at shorter times reduction in the weight of gap tuned frequency components." Could you please further explain the meaning of the gap tuned frequency components? And why are they special?

We have now added additional panels of laser pulse information to the phase diagram figures – in which context this "reduction in the weight of gap tuned frequency components" was mentioned – that hopefully serve to clarify this phrase. The improved figure is reproduced above as Fig. 8 of this reply (Fig. 6 of the revised manuscript). In these phase diagrams we have now added three points, labeled (A), (B) and (C) chosen to represent the three distinct laser pulse regimes: multi-cycle, single cycle, and very short time sub-cycle.

In the latter short time limit, case (A), a Fourier transform of the laser pulse, presented in panel (m), shows a reduction in weight at (and above) the gap, as compared to the other two cases, panels (g) and (j). As light-matter coupling is primarily determined by the strength of the laser pulse at the relevant gap (or above) frequencies – this determines the probability for a transition from valence band into the conduction band – the reduction in the weight of the Fourier transform at and above the gap indicates a significantly reduced possibility for charge excitation by the light pulses. This corresponds exactly to the data presented in the charge excitation phase diagram, panel (a) of Fig. 8 above, which shows that at very short sub-cycle times the charge excitation is significantly reduced.

We have expanded the discussion of the phase diagrams, lines 198-208 of the manuscript, and, along with the addition of representative laser pulse information to Fig. 6 of the manuscript, we hope this makes clear the meaning of the phrase referred to by the Referee. Our original formulation was definitely unclear, and we thank the Referee for pointing this out.

Third, in my understanding, the ultrafast dynamic refers to the light with high frequency. Hence, how does such ultrafast light generate valley charge and then the valley current in the biased bilayer graphene, as the band gap of the biased bilayer graphene is rather small?

For bilayer graphene the gap is – depending on the applied interlayer field – of the order of up to 200-300 meV. This corresponds to the mid infrared light spectrum, with a period of oscillation of the order of 10 fs. As the Referee indicates, this precludes the 1-2 fs control of valley current offered by dichalcogenides with gaps in the optical range, but nevertheless is still in the ultrafast femtosecond time domain.

Reviewer #3 (Remarks to the Author):

This paper studies the coupling of circularly polarized light to valley current in two dimensional semiconductors. They show that in the few cycle limit of light, the direction and magnitude of valley current can be controlled by the orientation and magnitude of light. They find that the effect exists both in visible spectrum gaps and infrared gaps.

The idea of creating valley current by light seems interesting and adds new results in the whole valleytronic panorama. The authors support their idea by introducing the concept of momentum space valley dipole and using the density matrix technique.

However, prior to publications, the authors should deal with some modifications of their current manuscript. I address here below my requests:

We thank the Referee for their very careful reading of the manuscript and the several questions raised that have prompted, we believe, several key improvements in our work. In particular:

1. We now include a treatment of excitons via *ab-initio* time-dependent density functional theory (TD-DFT). These calculations reveal that the few cycle symmetry breaking regime of circularly polarized light – characterized by a pure valley current that is CEP controllable – is (i) found in a realistic treatment of the problem via *ab-initio* calculation of WSe_2 and (ii) is robust to the inclusion of excitonic effects which interestingly enhance both the excited free carriers as well as the emergent CEP controllable valley current (due to exciton ionization by the short time strong field pulse). We believe that demonstrating that the physics identified on the basis of tight-binding models is found in realistic simulation considerably strengthens the case our paper makes, and we thank the Referee for this suggestion.

2. We have endeavored to improve the clarity of the physical picture presented for the underlying symmetry breaking mechanism.

3. The Referees interesting remark that few cycle pulses of circularly polarized light might separately control valley and charge current we can confirm, and have added this to our discussion of the physics of the symmetry breaking regime (lines 191-195).

1) In Reference 14, the authors have proposed THz induced giant spin and valley current. What are the differences of results between this manuscript and reference 14? The authors should address the concern both in the reply and revised manuscript.

The key difference between these two approaches is outlined in Fig. 9 below; we have also added text to both the introduction (lines 37-38) and discussion (lines 242-244) to clearly delineate these two methods. We thank the Referee for suggesting this clarification of what we believe are two quite distinct examples of light-matter coupling, as we explain below.

The hencomb pulse operates essentially as a hybrid in which the pulse components address quite distinct aspects of light-matter coupling: a sub-gap linearly polarized component evolves crystal quasi-momenta (*intra*-band transitions), while a gap tuned component excites *inter*-band transitions. Combined judiciously these act to both excite and displace charge from a high symmetry point – thus generating current. This can be

seen in Fig. 9a of the reply. In contrast the “K-pole” mechanism described in the present manuscript is not a pulse hybrid, but consists simply of the short time few cycle limit of a standard pulse: circularly polarized light. This limit we show exhibits emergent light-matter coupling physics with, notably, the ultrafast generation of a pure valley current.

Fig. 9: Hencomb, (a) and symmetry breaking “K-pole” (b-e) mechanisms to create current. In the hencomb mechanism a hybrid pulse, (f), consists of quite distinct components, a sub-gap tuned linearly polarized component and a gap-tuned circularly polarized component; together act to displace charge from the high symmetry K point. The necessity for sub-gap tuning of the linearly polarized component precludes few fs hencomb pulses. In contrast the “K-pole” mechanism explored in this work is not a hybrid – rather it is the short time limit of a pure circularly polarized pulse, (g), that, we show, manifests new emergent physics in this limit – notably the ability to generate current. In contrast to the hencomb this allows (i) significant reduction in the pulse duration and (ii) does not displace charge from the high symmetry valley centre, but rather breaks symmetry at the valley centre by the appearance of the dipole-like component to the excited state population.

These distinct mechanisms entail very different physics of light-matter coupling for the two pulses. In particular the time domain of the hencomb pulse, due to the requirement to have a sub-gap linearly polarized component, cannot act at the ultrafast few femtosecond time scales of the “K-pole” mechanism described in the present manuscript. We further illustrate the distinction by contrasting two possible applications, one possible with the hencomb pulse and one with the few cycle limit of circularly polarized light discussed in the present manuscript.

1. For the hencomb pulse the charge excitation retains C_3 symmetry about its “centre of mass”, but this centre is displaced from the high symmetry K point by the partner linearly polarized component. In contrast, in the present work the “centre of mass” of the excitation remains at the K point but suffers symmetry lowering in the few cycle limit. This raises the possibility that the hencomb pulse may offer a route to creating an excitonic current (if all conduction band states are significantly displaced from K, then the excitonic wavefunction, constructed from them, must also have overall finite momentum), a feature that seems less likely on the basis of the short time limit of circularly polarized light explore in the present manuscript.

3. In the present work the symmetry lowering mechanism occurs due to the interplay between the valley Berry curvature and orientable few cycle pulse trajectories. We believe

that in complex multi-band contexts, in which the Berry curvature will be band dependent, then this “K-pole” approach may then offers the exciting possibility of a new route towards band selective excitation by light, and effect that would not be possible with the hencomb pulse.

Fig. 10: Schematic of the three k-vectors – the C_3 star – included in the sum for the $D(\mathbf{k})$ function, Eq. 4.

2) The expression of density $D(\mathbf{k})$ from Eq. (1) is confusing. It seems that there are only two terms in the summation. How does it preserve C_3 symmetry? The authors should provide a schematic plot to clarify the expression.

Unfortunately there was a typo in the original formula, which gave the summation the appearance of two terms. The correct formula is

$$D(\mathbf{k}) = |c_{\mathbf{k}}|^2 - \frac{1}{3} \sum_{i=1}^3 |c_{M_i \mathbf{k}}|^2 \quad (\text{Equation 4})$$

which makes clear that the sum is over three terms. The meaning of this formula is simply that from the conduction band occupation at \mathbf{k} , the first term in Eq. 4, is then subtracted the star average, i.e. the average over the three C_3 related vectors $\mathbf{k}_i = M_i \mathbf{k}$. This function is therefore zero everywhere if every set of C_3 related k-vectors have the same conduction band occupation i.e. the total laser induced charge excitation respects the valley C_3 symmetry. Non-zero values of $D(\mathbf{k})$, on the other hand, indicate a lowering from the native C_3 valley symmetry. We thank the Referee for pointing out this typo.

3) The authors discuss the long time multi cycle and ultrashort single cycle limit in Figure 2, respectively. What about the middle regime? The authors should provide data to show the evolution of physical properties from ultrashort to long time cycle limit.

Fig. 11: Dependence of (a) the current and (b) the maximum value of the absolute value of $D(\mathbf{k})$, the symmetry breaking density, for times that interpolate between the long time multi-cycle high symmetry regime and the short time few cycle symmetry breaking regime. The current is seen to fall continuously to zero, behaviour in close correspondence with $\max |D(\mathbf{k})|$, indicating that this symmetry breaking density is correlated to the short time emergence of a valley current generated by circularly polarized light.

We present this data above as Fig. 11 of the reply, in which we interpolate between the single cycle pulse (full width half maxima ~ 2 fs) and the long time pulse (duration > 10 fs). In panel (a) is shown the valley current characteristic of the symmetry breaking regime that emerges at short times, while in panel (b) we present the maximum of $|D(\mathbf{k})|$.

As can be seen in panel (a) the emergent valley current found for the few cycle limit continuously and rapidly drops as the duration of the pulse is increased. Interestingly, the close correspondence found between this current and the maximum of $|D(\mathbf{k})|$, indicates again the key role the symmetry breaking density $D(\mathbf{k})$ plays in the ultrafast limit of light-valley coupling physics. This figure we have now included in the Supplemental as a complementary point of view to the presentation of this data implicitly in the phase diagrams, and we thank the Referee for suggesting this addition.

4) In Figure 2 (d)-(f), C_3 symmetry is broken. Is there any intuitive physical understanding of the symmetry breaking?

We believe that our explanation of this in the previous version of our manuscript was not sufficiently clear and have endeavored to improve the clarity of our discussion of the underlying physical effect, with the addition of a new figure, Fig. 3 of the revised manuscript, reproduced below as Fig. 13 of this reply. The appropriate section of the manuscript has been heavily revised (lines 104-131) and we now hope that our explanation of the underlying physics of the symmetry breaking, and the central role of the Berry curvature and the light induced evolution of momentum $\mathbf{k}(t)$ is now much clearer.

In short summary, the essential difference between the few cycle limit and multi-cycle pulses is that the dynamical trajectories in momentum space for the former case consist of one single loop – see Fig. 13d of the reply -- and must therefore evolve either into or out of regions of high Berry curvature, with consequent difference in excited charge and symmetry breaking, while in the multi-cycle limit these two processes average out leading to no difference of excited charge, preserving C_3 symmetry. We now describe this difference in detail.

For multicycle pulses the trajectory $\mathbf{k}(t)$ induced by light describes circular motion in momentum space. For two initial momenta $(0, \pm k_y)$, indicated by crosses in Fig. 13a of this reply, these trajectories approximately map into each other under $k_y \rightarrow -k_y$ and, as the Berry curvature also respects $k_y \rightarrow -k_y$, the curvature “recorded” by each trajectory will thus be very similar. This is shown in Fig. 13b in which the Berry curvature $\Omega(\mathbf{k}(t))$ is plotted as a function of time for each initial momenta. The oscillation results from the two

initial momenta $(0, \pm k_y)$ successively evolving closer to the region of high Berry curvature at K, and subsequently away from it, as a result of the circular motion entailed by $\mathbf{k}(t)$, while the $\pi/2$ phase shift has its origin in the fact that while the $(0, +k_y)$ trajectory evolves towards K the $(0, -k_y)$ trajectory necessarily evolves away from K. Regions of high Berry curvature are associated with excitation of charge to the conduction band, however over many cycles the effect for the two initial momenta $(0, \pm k_y)$ averages out and, as a result and as may be seen in Fig. 13c, the conduction band occupations post laser pulse are nearly identical, $|c_{+k_y}|^2 \sim |c_{-k_y}|^2$.

Fig. 13: *The Physical mechanism of emergent short time symmetry breaking.* (a) For long time pulses, the laser driven $\mathbf{k}(t)$ -trajectories of two initial momenta $(0, +k_y)$ and $(0, -k_y)$, indicated by the crosses in panel (a), approximately map into each other under $k_y \rightarrow -k_y$ and, as a result, the Berry curvature "recorded" along the dynamical trajectories exhibit similar but phase shifted oscillations, panel (b), generating ultimately nearly identical final conduction band occupations $|c_{\mathbf{k}}|^2$, panel (c). Reduction in pulse duration to $\sim 1-2$ fs drives light-matter coupling to a symmetry breaking regime, (d-f), in which the few cycle laser driven trajectories for initial momenta $(0, \pm k_y)$ now lack $k_y \rightarrow -k_y$ reflection symmetry, panel (d), with, as a consequence, very different Berry curvatures "recorded" along the two dynamical trajectories, panel (e). The $(0, +k_y)$ initial momenta – whose dynamical $\mathbf{k}(t)$ trajectory drives it *into* the region of high Berry curvature near the K point – has increased conduction band charge $|c_{\mathbf{k}}|^2$ as compared to the $(0, -k_y)$ initial momenta whose dynamical trajectory drives it *away* from the region of high Berry curvature, panel (f). The laser induced charge excitation inherits this $k_y \rightarrow -k_y$ symmetry breaking, leading to the finite "K-pole" structure exhibited by momentum resolved excited state population shown in Fig. 2(c,d) of the manuscript.

For the single cycle pulse the situation is very different. Now the $\mathbf{k}(t)$ trajectory executes essentially only one loop, which may either be *into* the region of high Berry curvature, or *away* from it, see Fig. 13d. For the $(0, +k_y)$ initial momenta, light evolution of momenta into high $\Omega(\mathbf{k})$ will result in increased excitation to the conduction band as compared to the $(0, -k_y)$ initial condition, which evolves away from the region of high Berry curvature. This can be seen in Fig. 13e,f. We now have $|c_{+k_y}|^2 > |c_{-k_y}|^2$ and thus the near $k_y \rightarrow -k_y$ equality of the conduction band occupations found for the multi-cycle case is broken.

As this will occur for every such $(0, \pm k_y)$ pair, the laser induced excited charge will inherit this symmetry breaking, exactly registered by the symmetry breaking density $D(\mathbf{k})$ for the few cycle pulse, Fig. 2(c,d) of the manuscript, but is absent for the multicycle pulse Fig. 2(a,b) of the manuscript.

We thank the Referee for pointing out the unclear presentation of our explanation for the underlying physics of the few cycle symmetry breaking regime of circularly polarized light, and hope that our extensive revision of this section of the manuscript has resulted in a clearer presentation.

5) In Figure 3, authors mention that the orientation can be controlled by the pulse carrier envelope phase (CEP). How to tune CEP experimentally? The authors should provide more discussions on the experimental realization of the effect in the manuscript.

We have now referred to two key papers, Ref. 28 and 29 of the revised manuscript, in the discussion section (lines 247-249) at the point at which we consider the experimental requirements for investigating the symmetry breaking few cycle regime of circularly polarized light. The tuning and control of carrier envelope phase at the few cycle limit in fact already exists, and thus experiments probing of the physics we describe are possible and hopefully will be carried out.

6) In line 149-151, there is a statement "at longer times the multi cycle pulses lack orient ability, while at shorter times reduction in the weight of gap tuned frequency components, as well as loss of curvature of the pulse trajectory in momentum space". The authors should provide more details to explain the statement.

We have now expanded the text at this point (lines 197-208), but also to clarify this added a series of panels to the phase diagram figure (the context for this statement) in which we show explicitly the pulse properties for the three regimes of light: multi-cycle, single cycle, and very short time sub-cycle.

In the latter short time limit, case (A), see Fig. 6 of the revised paper included also as Fig. 14 of the reply below, a Fourier transform of the laser pulse, presented in panel (m), shows a reduction in weight at (and above) the gap, as compared to the other two cases, panels (g) and (j). As light-matter coupling is primarily determined by the strength of the laser pulse at the relevant gap (or above) frequencies – this determines the probability for a transition from valence band into the conduction band – the reduction in the weight of the Fourier transform at and above the gap indicates a significantly reduced possibility for charge excitation by the light pulses. Note that this corresponds exactly to the data presented the charge excitation phase diagram, panel (a), which show that at very short times the charge excitation is significantly reduced.

Fig. 14: Phase diagrams of the symmetry breaking regime of light-valley coupling. Presented are the excited charge and its valley polarization, panels (a,b), and the excited current and its valley polarization, panels (c,d), plotted as a function of pulse full width half maxima (FWHM) and central frequency. The emergence of valley current signals valley symmetry breaking, panel (c), and occurs at times corresponding to a gap tuned single cycle pulse, \hbar/Δ , ~ 1 -2 fs for a gap of 1.6 eV, driven by an orientable trajectory in momentum space. At longer time multi-cycle pulses this property is lost and the valley current correspondingly vanishes. While current persists into the sub-single-cycle attosecond regime, compare (c,d), valley polarization is lost, establishing a temporal limit on the time scale for which valley current can be generated. These distinct regimes of light-matter coupling are highlighted at the points indicated in the phase diagrams: multi-cycle, single cycle, and very short time sub-cycle. Panels (e-m) display for each of these cases the parametric $\mathbf{A}(t)$ -trajectory, the pulse vector potential, and its Fourier transform, revealing that each light-matter coupling regime to be associated with a distinct pulse topology.

Each of these regimes – A, B and C – is also, as can be seen from Fig. 14 above, endowed with a distinct pulse topology. For the multicycle pulse the parametric trajectory, case A panel (e), is approximately circular, which upon reduction of duration first shows a dramatic lowering of mirror symmetry as it becomes effectively a single loop, case B panel (h), before eventually limiting to a nearly linearly polarized pulse, case C panel (k).

This ultrashort pulse pulse, due to the approach to linear polarization, shows sharply reduced valley polarized of light induced charge; this can be seen by comparing the point denoted C denoted in Fig. 14b (charge valley polarization) and Fig. 14d (current valley polarization), with those denoted A and B.

We hope that with the addition of extra panels to Fig. 5 of the manuscript (Fig. 4 of the original manuscript), along with clarification of the text, then the meaning of our original statement is now unambiguous and thank the Referee for prompting this revision of what was definitely an unclear presentation.

7) According to Figure 1, it seems that valley current is accompanied by valley charge. Is it possible to realize the valley current without valley charge, that is, independent control of valley charge and valley current?

The Referee raises a very interesting point, and indeed the answer is in the affirmative: it is possible in the ultrafast few cycle regime to control valley charge and current independently by laser light pulse parameters. To see this note the points denoted (B) and

(A) in Fig. 14a of this reply (Fig. 6 of the revised manuscript) that presents the excited charge as function of full width half maxima (FWHM) and frequency, and Fig. 14c that presents excited current as a function of FWHM and frequency. Point (B) falls in a region of strong valley current but weak charge excitation, while point (A) falls in a region of weak valley current but significant charge excitation. Thus by variation of the FWHM and frequency one can tune to have distinct values of the valley current and charge. We thank the Referee for pointing this out and have added this point to our discussion of the parameter space in while the “K-pole” symmetry breaking mechanism emerges (lines 191-195).

8) In the manuscript, circularly polarized light is considered. What will happen if we apply the linearly polarized light?

We have recently considered this case [Nano Lett. 23 11533–11539 (2023)]. The situation is now very different as with linear light charge excitation is no longer valley selective. However, by combining two opposing effects – the tendency for valley trigonal warping to produce opposite currents at the two conjugate valleys, and light asymmetry via zero carrier envelope pulses to produce identical currents at the two conjugate valleys – one can tune a light pulse to achieve pure valley currents. In this case, however, control over valley charge is not possible: the excited charge at both valleys is identical in all cases due to the linear polarization of the light pulse. The linearly polarized case thus presents a quite distinct mechanism of short time valley physics, relying on *material asymmetry* (the valley trigonal warping) and not *light asymmetry* as for the case of the few cycle limit of circularly polarized light.

9) In the section of Methods, there is a system named gapped graphene. However, for monolayer gapped graphene, the gap value can not be as large as 1.6 eV. The authors should change the name.

We have changed the name to “avoided crossing model” and indicated that this is not intended to model a “modified” graphene, such as may be obtained from doping or substrate effects, but rather provides a minimal model to discuss substantially gapped systems endowed with valley Berry curvature. We have done this throughout the text and also at the appropriate point in the Methods section, lines 263-265.

10) In the studies of light-induced valley charge, excitons and trions are formed as a result of Coulomb interaction. How does such interaction affect the physics proposed in the manuscript?

To explore the role of excitonic physics we have performed time-dependent density functional theory (TD-DFT) calculations of WSe_2 employing a recently proposed Kohn-Sham-Proca method [arXiv:2401.16140], shown to capture both the bound excitonic peaks and spectral weight renormalization in the linear response regime, but also key features of non-equilibrium excitonic physics such as the laser pulse induced “bleaching” (reduction in absorption) of the excitonic peak and the appearance of excitonic side bands. While we cannot distinguish exciton and trion formation in the dynamics, the method does treat excitonic and free carrier physics on an equal footing, making this the ideal tool for investigating the behaviour of excitons under pump laser conditions in which both free carriers and bound states such as excitons will play equally important roles.

Our findings are shown in Fig. 15 of this reply (now Fig. 5 of the manuscript). The linear response absorption spectrum of WSe_2 , Fig. 15a, is seen to be very well captured: the excitonic peak at 1.66 eV is reproduced in excellent agreement with experiment, but also the position of a second satellite excitonic peak is captured (though not its amplitude). Application of a single cycle pump laser pulse generates in both cases a strong current response, Fig. 15b. Interestingly, the current response is greater in magnitude with the inclusion of excitonic physics; as shown in the Supplemental via the time dependent density of states this occurs as the strong field pulse can ionize excitons leading to an increase in the free carrier population. This increase in free carrier population is also discussed in a recent paper [arXiv:2307.14693], and is indicative of an intrinsic coupling of exciton and free carrier populations under pump laser conditions.

The critical test of robustness of the mechanism to excitonic effects is whether this current is CEP controllable – the hallmark of the orientable trajectory in momentum space and thus of the symmetry breaking regime. As can be seen in Fig. 15(c,d), this clearly so: the sinusoidal oscillation found in the minimal tight-binding avoided crossing model is reproduced by the TD-DFT current including excitonic effects. We thus conclude that the effect we report is robust to the inclusion of excitonic effects.

Fig. 15: Impact of excitonic physics on the symmetry breaking regime, example of WSe_2 : The experimental optical absorption for WSe_2 , presented alongside that calculated via time-dependent density functional theory (TD-DFT) and by the random phase approximation (RPA). The TD-DFT calculation reproduces very well the pronounced bound exciton at 1.66 eV, capturing also the position of the excitonic satellite (both indicated by vertical lines). Note that we scissors correct the local density approximation (LDA) gap of 1.44 eV to the 1.98 eV experimental gap. (b) The action of the single cycle symmetry breaking laser pulse, similar to that presented in Fig. 1(f) of the manuscript but with the frequency scaled to the 1.44 eV LDA gap, reveals an increased current response upon the inclusion of excitonic effects in the TD-DFT calculation. Remarkably, the control over the current direction by lightform the carrier envelope phase - the hall mark of the "K-pole" symmetry breaking regime of valley physics -- is robust to the inclusion of excitons in the dynamics.

We thank the Referee for pointing out the importance of including excitons in our analysis: we believe that by demonstrating that *ab-initio* calculation, treating excitons and free carriers on an equal footing in highly non-equilibrium pump laser conditions, also reveals key characteristics of the ultrafast symmetry breaking regime that we have identified through simple models considerably strengths the case our paper makes.

REVIEWER COMMENTS

Reviewer #1 (Remarks to the Author):

The authors expanded their discussion about the generation of valley currents. However, it seems to me that the provided explanation might not be correct, or at least not sufficiently proven:

The authors claim that the valley current "has its origin in the dependence of the dynamics of the Berry curvature on the 'shape' of the light induced evolution of crystal momentum" (line 45). I understand the observation they make and elaborate on in Figure 3 and its discussion. However, the explanation is based on the statement that "Regions of high Berry curvature are associated with excitation of charge to the conduction band" (line 116). I see the correlation that the authors point out, but the manuscript provides neither arguments nor references for a causal connection between the Berry curvature and the excitation of charges, which is the foundation of the authors' explanation of the observed valley current. In fact, the excitation of charges (and their transition probabilities) is typically connected to the k-dependent dipole matrix elements or the gap energy via the Keldysh parameter, depending on the excitation conditions (see for example <https://arxiv.org/pdf/1502.02180.pdf>), which the authors do not address. At the same time, I am not aware of a physical connection between the Berry curvature and the transition probability as the authors claim. If such a connection exists, the authors should be able to show this analytically from their model. But the observed correlation does not imply causation.

The issue described above is the most critical to me as it concerns the explanation of the main result of the paper. Beyond that, I have some further comments:

1) The authors try to make the point that the wave form is "characterized by a scalar" and should therefore only couple to "scalar degrees of freedom" (line 32-33) and describe this as "common sense" (line 41). I would argue against this. First, the polarization is an inherently vectorial quantity and cannot be described by a scalar. Second, it is established that few-cycle pulses can generate CEP-dependent currents (for example <https://www.nature.com/articles/nature11567>). Third, to me it seems obvious that complex wave forms cannot simply be described by a scalar. Therefore, the claimed "conundrum" (line 132) does not exist in my opinion and seems constructed.

2) Line 152 defines the "K-pole" and the authors note a "remarkable" proportionality between the "K-pole" and the valley current. However, observing that $\nabla_k E_i$ in Eq. (14) is proportional to k near $k=K$ makes this a trivial result. I find it unfortunate that simple connections like this are not discussed (which could improve the clarity of the paper) and instead oversold or overlooked.

3) I find parts of the supplement difficult to follow. Some of the symbols are not defined (for example $\epsilon_{i,k}(t)$ in Eq. (5)) and some of the notation is confusing (for example when the k dependence of variables is omitted such as for E and ϵ in Eq. (8), or when vector quantities do not follow the convention of using bold font such as a_i in Eq. (18)). Section 1 and 2 seem to introduce similar equations but using different notations, which I find confusing. For example, Section 1 uses momenta $H(p)$ while Section 2 uses wave vectors $H(k)$. Section 1 only considers macroscopic currents $j(t)$ while Section 2 introduces microscopic currents $j_k(t)$ without connecting them to the macroscopic currents in Section 1.

Due to the remarks above I would still not recommend the manuscript for publication in its current form.

Reviewer #2 (Remarks to the Author):

After carefully reading the responses, I think all the comments are well addressed by the authors, and the manuscript is revised accordingly. So, I recommend its publication in its current form.

Reviewer #3 (Remarks to the Author):

The authors add new data which include the effect of excitons, and their other revisions have improved the clarity of the paper. They have completely addressed all of my comments. I recommend acceptance for publication in Nature Communications.

Referee 1

We thank the Referee for the careful second reading of our manuscript. In particular the Referee suggests discussing interband transitions in terms of the interband dipole matrix elements $d(k)$; from the analytical forms of $d(k)$ and the Berry curvature $\Omega(k)$ we find that the arguments of the paper can be reformulated without change in terms of $d(k)$ instead of $\Omega(k)$. We appreciate this suggestion by the Referee. All further questions raised by the Referee are addressed below in detail.

The authors expanded their discussion about the generation of valley currents. However, it seems to me that the provided explanation might not be correct, or at least not sufficiently proven: The authors claim that the valley current "has its origin in the dependence of the dynamics of the Berry curvature on the 'shape' of the light induced evolution of crystal momentum" (line 45). I understand the observation they make and elaborate on in Figure 3 and its discussion. However, the explanation is based on the statement that "Regions of high Berry curvature are associated with excitation of charge to the conduction band" (line 116). I see the correlation that the authors point out, but the manuscript provides neither arguments nor references for a causal connection between the Berry curvature and the excitation of charges, which is the foundation of the authors' explanation of the observed valley current. In fact, the excitation of charges (and their transition probabilities) is typically connected to the k -dependent dipole matrix elements or the gap energy via the Keldysh parameter, depending on the excitation conditions (see for example <https://arxiv.org/pdf/1502.02180.pdf>), which the authors do not address. At the same time, I am not aware of a physical connection between the Berry curvature and the transition probability as the authors claim. If such a connection exists, the authors should be able to show this analytically from their model. But the observed correlation does not imply causation.

The Referee has identified an interesting question: the role (if any) of the Berry curvature in generating transitions in the ultrafast regime. We have reflected on this question and feel that while light-valley coupling can be couched in terms of Berry curvatures in the wave-packet picture¹, for our explanation the k -dependence of the dipole matrix elements is more suitable. As we now show the arguments of the paper can be recast in terms of the latter without change.

We firstly note that the Keldysh parameter is $\sim 10^{-3}$ for our pulses, indicating, as expected, a highly non-perturbative regime with transitions controlled by the interband dipole matrix elements. To describe the dynamics these terms one deploys Houston type states, allowing the Schrodinger equation $i\partial_t\Psi(\mathbf{k}) = H\Psi(\mathbf{k})$ to be rewritten involving only the interband dipole matrix elements (the *intra*band dipole matrix elements, i.e. the Berry connection, can be removed by a gauge transform):

$$\partial_t a = \mathbf{E}(t) \cdot \mathbf{d} a \quad (\text{Eq 1})$$

where

$$\mathbf{d}_{nm} = (1 - \delta_{nm}) \langle \phi_{n\mathbf{k}} | \nabla_{\mathbf{k}} | \phi_{m\mathbf{k}} \rangle \quad (\text{Eq 2})$$

and a are the expansion coefficients of $\Psi(\mathbf{k})$ in terms of the Houston states $|\phi_{n\mathbf{k}}\rangle$ with $\mathbf{E}(t)$ the electric field.

For an avoided crossing model we thus have a single k -dependent dipole matrix element to consider. Labeling the valence (conduction) bands by - (+) we can find this, in the Dirac-Weyl approximation, analytically as

$$\mathbf{d}_{-+}(\mathbf{k}) = -i\frac{\hbar v_F \Delta}{2\epsilon^2} \hat{\mathbf{k}} + \frac{\nu \hbar v_F}{2\epsilon} \hat{\phi} \quad (\text{Eq 3})$$

where Δ is the gap parameter (the gap is twice this), v_F the Fermi velocity, and $\nu = \pm 1$ the valley index. Note this result is expressed in polar coordinates. This can be compared to the Berry curvature, which in the same model is given by

$$\Omega_\sigma(\mathbf{k}) = \frac{-\sigma \nu (\hbar v_F)^2 \Delta}{2\epsilon^3} \quad (\text{Eq 4})$$

where $\sigma = \pm 1$ labels the bands. In both of these equations $\epsilon = \sqrt{\Delta^2 + (\hbar v_F k)^2}$ is conduction band energy.

The dipole matrix elements thus share two crucial features with the Berry curvature: (i) they decay with energy ϵ away from the high symmetry K point, and, (ii) the K point is the momentum of maximum interband excitation. In Fig. 1 of this reply we compare the normalized curvature with the normalized dipole matrix elements, revealing, as expected, a very similar form in momentum space. The argument of the paper, that “the Berry curvature recorded along a trajectory” distinguishes loops of momentum evolution that either evolve into the region of high Berry curvature or away from it, can thus, from Eqs. 1 and 3, immediately (and more rigorously) be recast in terms of dipole matrix elements.

We have thus altered the manuscript removing references to Berry curvature and replacing by interband dipole matrix element. A discussion of the role of dipole matrix elements has been added to the SI in which the formulas above are carefully derived (Sec. 2 of the revised SI), with appropriate linkage in the manuscript (lines 122-123). We have further reformulated the derivation of interband current for the avoided crossing model entirely in terms of dipole matrix element, which also leads to a simplifying of this discussion (Sec. 3 of the revised SI).

Figure 1: (a) The \mathbf{k} -dependence of the Berry curvature along with two curves indicating the evolution of momentum for two \mathbf{k} -points related by $k_y \rightarrow -k_y$. (b) The \mathbf{k} -dependence of the normalized magnitude of the interband dipole matrix element, $d(k)/d(0)$, with the same two curves. The argument underpinning to short time symmetry breaking of circularly polarized light, i.e. that dynamical trajectories originating from k_y and $-k_y$ are distinguished, as the former evolves momentum into a region of high interband transition while the latter away from it, thus holds both when couched in terms of the Berry curvature or the interband dipole matrix elements $d(k)$ as underpinning transitions. As can be seen from Eq. 1, this is most clearly stated in terms of the latter.

1) The authors try to make the point that the wave form is "characterized by a scalar" and should therefore only couple to "scalar degrees of freedom" (line 32-33) and describe this as "common sense" (line 41). I would argue against this. First, the polarization is an inherently vectorial quantity and cannot be described by a scalar. Second, it is established that few-cycle pulses can generate CEP-dependent currents (for example <https://www.nature.com/articles/nature11567>). Third, to me it seems obvious that complex wave forms cannot simply be described by a scalar. Therefore, the claimed "conundrum" (line 132) does not exist in my opinion and seems constructed.

While we of course agree with the Referee that polarization is vectorial and thus, in general, complex electromagnetic waveforms cannot be characterized by a scalar, we would argue that this is not true for "special case" waveforms. An example is circularly polarized light, in which the polarization vector rotates uniformly in time. Such pulses are then characterized not by the inherently vectorial polarization but by a *property of this vector*, the sign of its rotation in time, a scalar. This can be contrasted with the example of linearly polarized light in which the polarization vector, now fixed, allows the lightform to be characterized directly by its polarization vector. Each of these special cases, furthermore, associates with a striking light-matter phenomena: (i) the scalar valued helicity of circularly polarized light determines which valley charge is excited at, i.e. light-valley coupling, the foundational phenomena of valleytronics, while (ii) linearly polarized light, by having a fixed preferred direction, can in the few cycle limit break symmetry in momentum space and excite *charge current*²⁻⁶ (i.e. equal current flow at both valleys).

Thus while we agree with the Referee that it is well known that few cycle pulses can generate charge current with CEP dependence, all such cases, including that cited by the Referee, belong to category (ii) above, that of linearly polarized light pulses. The question then arises: can light be employed to controllably generate *pure valley current*? For this one evidently requires valley pure excitation, and thus a pulse of circularly polarized light. However circularly polarized light implies a rotating polarization vector, no preferred symmetry breaking direction in momentum space, and so no possibility to excite current. This is the conundrum: to generate ultrafast pure valley current one apparently requires the combination of two contradictory light pulses, in the sense of requiring a pulse both possessing both a fixed polarization vector to generate current and a rotating polarization vector to exclusively couple light to a single valley.

In this work we resolve this apparent conundrum by showing that at ultrashort times the electromagnetic vector potential of circularly polarized light becomes a "loop" possessing both vectorial character, as well as inheriting the crucial sign of rotation of the corresponding long time circularly polarized pulse. As we stress in the paper, this represents both an interesting short time "emergent" symmetry breaking, in which an ultrafast version of a light pulse has very different light-matter coupling from its long time counterpart, but also one that will, we predict, allow the generation of pure valley current at few femtosecond times.

In light of the Referee's comment we have modified the paper to (i) express the "conundrum" in terms of polarization vectors and to make clear that linearly polarized light can generate ultrafast charge (but not valley) current, citing both the example of the referee and others, lines 32-36, and (ii) the discussion of the "conundrum" is now couched both in terms of dynamical trajectories as well as the polarization vector, lines 137-146.

2) Line 152 defines the "K-pole" and the authors note a "remarkable" proportionality between the "K-pole" and the valley current. However, observing that $\nabla_k E_i$ in Eq. (14) is proportional to k near $k=K$ makes this a trivial result. I find it unfortunate that simple connections like this are not discussed (which could improve the clarity of the paper) and instead oversold or overlooked.

The key findings are the emergent vectorial character of circularly polarized light, manifested by the “K-pole”, and not, as the Referee notes, the relation between the analogue of the dipole moment of this “K-pole” and the current which is trivial. We have amended this section describing the control over pure valley current by few cycle circularly polarized pulses to make this clear, lines 159-161.

3) I find parts of the supplement difficult to follow. Some of the symbols are not defined (for example $\epsilon_{i,k(t)}$ in Eq. (5)) and some of the notation is confusing (for example when the k dependence of variables is omitted such as for E and ϵ in Eq. (8), or when vector quantities do not follow the convention of using bold font such as a_i in Eq. (18)). Section 1 and 2 seem to introduce similar equations but using different notations, which I find confusing. For example, Section 1 uses momenta $H(p)$ while Section 2 uses wave vectors $H(k)$. Section 1 only considers macroscopic currents $j(t)$ while Section 2 introduces microscopic currents $j_k(t)$ without connecting them to the macroscopic currents in Section 1.

We thank the Referee for identifying these typos and have fixed them; the k -dependence has now generally been indicated along with vectorial character by bold font (in the case of the a_i 's these are kets and we retain there the non-bold font). For the same of clarity of notation we have now split this content over three sections: (1) the general case (2) the avoided crossing model and its dipole matrix elements and (3) an analytical form of the interband current, employed to explain the early time rotation of the current direction away from the CEP angle.

- (1) Souza, I.; Vanderbilt, D. Dichroic f -Sum Rule and the Orbital Magnetization of Crystals. *Phys. Rev. B* **2008**, 77 (5), 054438. <https://doi.org/10.1103/PhysRevB.77.054438>.
- (2) Schiffrin, A.; Paasch-Colberg, T.; Karpowicz, N.; Apalkov, V.; Gerster, D.; Mühlbrandt, S.; Korbman, M.; Reichert, J.; Schultze, M.; Holzner, S.; Barth, J. V.; Kienberger, R.; Ernstorfer, R.; Yakovlev, V. S.; Stockman, M. I.; Krausz, F. Optical-Field-Induced Current in Dielectrics. *Nature* **2013**, 493 (7430), 70–74. <https://doi.org/10.1038/nature11567>.
- (3) Higuchi, T.; Heide, C.; Ullmann, K.; Weber, H. B.; Hommelhoff, P. Light-Field-Driven Currents in Graphene. *Nature* **2017**, 550 (7675), 224–228. <https://doi.org/10.1038/nature23900>.
- (4) Li, Q. Z.; Elliott, P.; Dewhurst, J. K.; Sharma, S.; Shallcross, S. Ab Initio Study of Ultrafast Charge Dynamics in Graphene. *Phys. Rev. B* **2021**, 103 (8), L081102. <https://doi.org/10.1103/PhysRevB.103.L081102>.
- (5) Sharma, S.; Dewhurst, J. K.; Shallcross, S. Light-Shaping of Valley States. *Nano Lett.* **2023**, 23 (24), 11533–11539. <https://doi.org/10.1021/acs.nanolett.3c03245>.
- (6) Motlagh, S. A. O.; Nematollahi, F.; Mitra, A.; Zafar, A. J.; Apalkov, V.; Stockman, M. I. Ultrafast Optical Currents in Gapped Graphene. *J. Phys.: Condens. Matter* **2019**, 32 (6), 065305. <https://doi.org/10.1088/1361-648X/ab4fc7>.

REVIEWERS' COMMENTS

Reviewer #1 (Remarks to the Author):

All comments have been satisfactorily addressed in the reply and changes to the manuscript. I recommend the article for publication.

Rereading my previous report, I hope the authors did not perceive the tone as too harsh. As a non-native speaker my focus on precise comments may have led to a less polite tone than intended.